# Decadal Evolution of Aerosol-Mediated Ozone Responses in Eastern China under Clean Air Actions and Carbon Neutrality Policies

Yasong Li[1,3], Chen Li[2], Yaoyu Li[1], Tijian Wang[3*], Mengmeng Li[3], Yawei Qu[4], Hao Wu[5], Min Xie[6], Yanjin Wang[1]

1.  *College of Environmental Economics, Henan Finance University, Zhengzhou, 450046, China*

2.  *School of Energy and Chemical Engineering, Tianjin Renai College, Tianjin, 301636, China*

3.  *School of Atmospheric Sciences, Nanjing University, Nanjing,210023, China*

4.  *College of Intelligent Science and Control Engineering, Jinling Institute of Technology, Nanjing,211112, China*

5.  *Key Laboratory of Transportation Meteorology of China Meteorological Administration, Nanjing Joint Institute for Atmospheric Sciences, Nanjing, China*

6.  *School of Environment, Nanjing Normal University, Nanjing 210023, China*

***Correspondence to:* Tijian Wang (tjwang@nju.edu.cn)**

**Abstract:**

Despite substantial reductions in $PM_{2.5}$ and other pollutants, ozone ($O_3$) in eastern China has increased over the past decade, yet the influence of aerosol processes—including aerosol–radiation interactions (ARI) and heterogeneous chemistry (HET)—on these trends remains insufficiently explored, particularly during Clean Air Action Plan (CAAP, Phase I: 2013–2017; Phase II: 2018–2020) and under carbon neutrality pathways. We applied a phase- and season-resolved WRF-Chem framework with explicit ARI and HET to quantify historical and projected $O_3$ changes in the Yangtze River Delta (YRD), linking aerosol effects with CAAP and carbon-neutrality pathways. We separate $O_3$ changes into those driven directly by anthropogenic emissions and meteorological variability, and those mediated by aerosol processes through ARI and HET. The results revealed that anthropogenic emissions and meteorological variability respectively dominated winter and summer $O_3$ increases. Winter $O_3$ increases were dominated by ARI: large aerosol reductions enhanced solar radiation, temperature, and photolysis, resulting in a photochemical $O_3$ rise (+1.14 (+0.74) ppb in Phase I (II)). Summer $O_3$ was more sensitive to HET. In Phase I, aerosol decreases weakened heterogeneous radical uptake, enhancing $O_3$ formation (+1.62 ppb). In Phase II, however, the net HET effect reversed sign (–2.86 ppb), driven by shifts in multiple heterogeneous pathways—including changes in radical uptake, HONO and $N_2O_5$ chemistry, and aerosol liquid water—rather than radical scavenging alone. Accounting for aerosol effects (AEs=ARI+HET), reductions in $PM_{2.5}$ and NOx increased $O_3$, while VOCs reductions consistently lowered $O_3$ in both seasons. Under carbon peaking and neutrality scenarios with AEs, winter $O_3$ increased by 6.7% and 10.7%, whereas summer $O_3$ decreased by 2.9% and 6.7%, highlighting seasonally contrasting responses. These results underscore the necessity of explicitly accounting for multi-path aerosol–$O_3$ interactions in both near-term air quality management and long-term climate mitigation to prevent unintended trade-offs and maximize co-benefits.

**Graphical Abstract:**

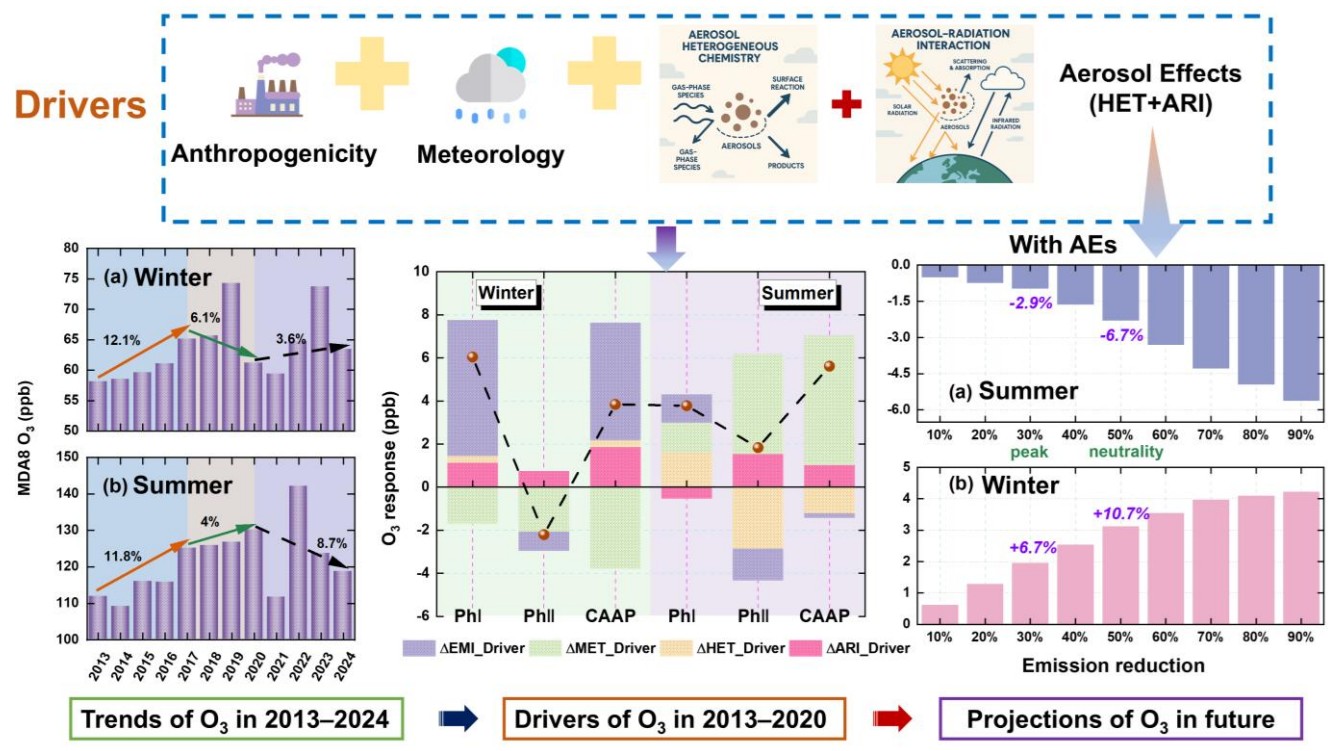

## 1. Introduction

Over the last decade, a series of landmark policy initiatives—such as the Air Pollution Prevention and Control Action Plan (Phase I: 2013–2017), the Three-Year Blue Sky Protection Campaign (Phase II: 2018–2020), and the subsequent dual-carbon strategy—have driven substantial and persistent declines in $PM_{2.5}$ concentrations across China's major urban clusters (Geng et al., 2024; Zhai et al., 2019). However, in sharp contrast to these successes, ground-level $O_3$ have continued to rise, particularly in economically developed regions such as Beijing-Tianjin-Hebei (BTH, (Zhao et al., 2023; Dai et al., 2023)), the Yangtze River Delta (YRD, (Li et al., 2023; Hu et al., 2025)), and the Pearl River Delta (PRD, (Chen et al., 2020)). For example, Yan et al. (2024) reported that the annual mean maximum daily 8-hour average (MDA8) $O_3$ in major Chinese cities increased from 106.0 μg m$^{-3}$ in 2013 to 131.1 μg m$^{-3}$ in 2022, with the most pronounced growth observed in the BTH and YRD regions. The emerging decoupling between $PM_{2.5}$ and $O_3$ trends underscores the growing complexity of air pollution control in China, suggesting that conventional precursor-oriented mitigation strategies may be insufficient to address secondary pollutants formed through nonlinear atmospheric processes. The increasing frequency and intensity of $O_3$ pollution episodes not only pose serious risks to human health and ecosystems (Liu et al., 2018; Li et al., 2020b) but also diminish the co-benefits of $PM_{2.5}$ mitigation. As China advances toward its goal of carbon neutrality, elucidating the mechanisms behind this counterintuitive $O_3$ rise has become both a scientific imperative and a policy priority.

Extensive research has identified anthropogenic emissions and meteorological variability as the two dominant drivers of observed $O_3$ increases (Ma et al., 2023a; Sun et al., 2019; Shao et al., 2024; Ni et al., 2024), particularly during the early stages of the CAAP. For instance, Dang et al. (2021) used the GEOS-Chem model to show that during the summer of 2012–2017, meteorological changes accounted for 49% of the $O_3$ increase in the BTH region and 84% in the YRD, while emission changes explained 39% and 13%, respectively. Recent efforts combining numerical modeling with machine learning have further highlighted the critical roles of solar radiation and temperature, especially during the COVID-19 lockdown. Zhang et al. (2025) attributed approximately 94% of the summer $O_3$ increase in the Hangzhou Bay area from 2019 to 2022 to meteorological influences, noting a growing dominance of meteorological drivers over emission-related factors. In addition, innovative metrics such as the $O_3$-specific emission–meteorology index (EMI/$O_3$) have been proposed to quantify these contributions, revealing that summer $O_3$ increases in cities like Beijing and Shanghai were largely governed by volatile organic compound (VOCs) emissions and meteorological shifts (Lu et al., 2025).

Beyond emissions and meteorology, aerosol effects (AEs) have emerged as important, though often overlooked, regulators of surface $O_3$. Aerosols influence $O_3$ formation through two principal mechanisms: aerosol–radiation interaction (ARI), which alter photolysis rates and boundary layer dynamics, and heterogeneous chemistry (HET), which removes hydroperoxyl ($HO_2$) radical and suppresses $O_3$ formation (Li et al., 2025; Li et al., 2024b; Li et al., 2019a; Gao et al., 2018). As aerosol loading has

substantially declined under clean air policies, the magnitudes and directions of these mechanisms may have shifted. For instance, Yu et al. (2019) found that reductions in $PM_{2.5}$ contributed to approximately 22% of the observed $O_3$ increase in the YRD during 2013–2017. Yang et al. (2024) quantified a 0.81 ppb increase in summer $O_3$ linked to the weakening of ARI under lower aerosol conditions. Previous analyses indicated that diminished aerosol modulation of photochemistry through ARI, photolysis rate suppression, and heterogeneous reactions jointly contributed to a 22.2%–57.3% enhancement in $O_3$ growth between 2014 and 2020 (Li et al., 2024a). Similarly, Liu et al. (2023a) identified weakened HET as the dominant mechanism behind $O_3$ increases across both phases of the CAAP. Moreover, precursor–$O_3$ relationships are strongly modulated by background aerosol levels, further emphasizing the need to assess $O_3$ responses under evolving aerosol conditions to ensure the effectiveness of co-control strategies. Anthropogenic emissions and meteorological variability act as external drivers that directly regulate precursor concentrations, atmospheric chemical regimes, and transport processes. In contrast, ARI and HET represent aerosol-mediated mechanisms that reshape the photochemical environment by altering photolysis rates and radical budgets. These aerosol-driven mechanisms determine the extent to which surface $O_3$ responds to precursor (particularly NOx) reductions or meteorological perturbations. This conceptual framework underpins our separation of $O_3$ changes into externally driven components and aerosol-modulated components in this study.

Despite increasing recognition of the role of aerosols in modulating surface $O_3$, several critical knowledge gaps remain. Most existing studies tend to isolate either ARI or HET rather than evaluate their combined and potentially synergistic effects. Additionally, few investigations adopt a phase- and season-resolved framework aligned with policy implementation timelines, and even fewer consider long-term projections under carbon neutrality pathways. Furthermore, the spatial heterogeneity and nonlinear chemical responses of $O_3$ under dynamic aerosol environments remain poorly characterized, particularly in densely populated and industrialized regions like the YRD. To address these gaps, this study employs an improved WRF-Chem modeling framework to conduct a comprehensive, phase-, season-, and mechanism-resolved assessment of AEs in the YRD from 2013 to 2024. By explicitly disentangling the effects of ARI and HET and integrating them with historical emission changes, meteorological variability, and future carbon neutrality–driven mitigation scenarios, we aim to systematically quantify the drivers of past $O_3$ trends and predict their future trajectories. Furthermore, we assessed the responses of $O_3$ to reductions in individual precursors ($PM_{2.5}$, NOx, VOCs, $NH_3$, and $SO_2$), thereby elucidating the conditions under which synergistic air quality and climate co-benefits can be most effectively realized. These results provide a scientific basis for the development of region-specific and seasonally adaptive $O_3$ mitigation strategies that are consistent with China's dual objectives of air pollution control and carbon neutrality.

## 2. Methodology

### 2.1 Model and dataset

To diagnose the mechanisms governing surface $O_3$ variability over eastern China under the Clean Air Action Plan (CAAP), we applied an improved configuration of the Weather Research and Forecasting model coupled with Chemistry (WRF-Chem, version 3.7.1, (Grell et al., 2005)). The analysis focused on two major implementation stages of the CAAP (Phase I and Phase II), with the objective of disentangling the relative contributions of emission controls, meteorological variability, and aerosol-mediated processes to long-term $O_3$ changes. Particular attention was devoted to two key aerosol effects (ARI and HET) and their roles in modulating $O_3$ trends. In addition, sensitivity experiments were conducted to quantify $O_3$ responses to precursor emission reductions, and to evaluate future surface $O_3$ behavior under carbon neutrality–oriented emission pathways while explicitly accounting for combined aerosol effects (ARI + HET). Building upon our previous modeling framework, the WRF-Chem setup largely followed configurations documented in earlier studies (Li et al., 2024a; Li et al., 2024b), with targeted enhancements to address the objectives of this work. A three-tier nested domain system was implemented, encompassing East Asia as the outermost domain, eastern China as the intermediate domain, and the YRD as the innermost domain (Figure S1). Biogenic emissions were calculated online using the Model of Emissions of Gases and Aerosols from Nature (Guenther et al., 2006). Numerical simulations were performed for January and July to characterize representative winter and summer conditions, respectively. Each seasonal simulation covered a five-week period (December 29 to February 1 for winter, and June 28 to August 1 for summer), with the initial three days excluded to allow for model spin-up and chemical equilibration. Beyond the seasonal analyses, the decadal evolution of maximum daily 8-hour average (MDA8) $O_3$ over the YRD during 2013–2024 was systematically examined for both seasons. Detailed information about the spatial distribution and technical characteristics of the monitoring stations and model configuration have been reported in our previous studies (Li et al., 2024a).

### 2.2 Aerosol effects enhancement

This work provided a comprehensive evaluation of aerosol-mediated influences on surface $O_3$ variability within the dual context of China's CAAP and prospective carbon neutrality pathways. Two representative aerosol-related processes (ARI and HET) were explicitly represented in the WRF-Chem modeling system to account for the coupled physical and chemical pathways through which aerosols regulate $O_3$ formation. The formulation, implementation, and performance evaluation of these processes followed the methodologies established in our earlier studies and are only briefly outlined here for completeness (Li et al., 2024b). Within this framework, ARI modulated $O_3$ concentrations through two primary mechanisms. First, aerosols attenuated incoming solar radiation, thereby influencing photolysis frequencies through light extinction. Second, aerosols perturbed meteorological conditions by altering radiative fluxes, giving rise to aerosol–radiation feedbacks (ARF). While ARF was natively supported in the standard WRF-Chem configuration, the default Fast-J photolysis scheme did not

dynamically account for aerosol optical properties, which led to the omission of aerosol extinction effects on photolysis rates. To overcome this deficiency, a customized coupling interface was implemented to link prognostic aerosol optical parameters—such as scattering and absorption coefficients—to the Fast-J module. This modification allowed aerosol optical depth to be calculated online and enabled photolysis rates to respond consistently to the evolving spatial and temporal distributions of aerosols.

Heterogeneous chemistry exerts complex influences on $O_3$ formation by altering radical budgets, modifying reactive nitrogen cycling, and changing aerosol-phase reaction rates. In the enhanced WRF-Chem, HET is represented through multiple pathways on dust and black carbon surfaces, including (1) heterogeneous uptake of $HO_2$, $OH$, $NO_2$, and $NO_3$; (2) nighttime $N_2O_5$ hydrolysis to $2HNO_3$; (3) heterogeneous formation of HONO from $NO_2$ uptake on carbonaceous aerosols; (4) $SO_2$ and $H_2SO_4$ heterogeneous oxidation; and (5) direct $O_3$ uptake on dust and black carbon surfaces. These processes collectively modify photolysis-driven radical initiation and NOx partitioning. Therefore, the net HET effect reflects the balance among several aerosol-mediated pathways rather than a single mechanism. The heterogeneous reactions considered in this study, together with their corresponding uptake coefficients ($\gamma$), were summarized in Table S1. Key parameters, including uptake coefficients, aerosol surface area densities, and photolysis scaling factors, followed values that had been validated in our previous modeling studies (Li et al., 2024b). The enhanced WRF-Chem system had been systematically assessed in earlier work and was demonstrated to realistically reproduce meteorological fields, aerosol characteristics, and trace gas concentrations in China, with particularly robust performance in YRD (Qu et al., 2023; Li et al., 2018).

**2.3  Numerical experimental designs**

To disentangle the respective and combined influences of anthropogenic emission changes, meteorological variability, and aerosol-related processes on surface $O_3$, three groups of numerical experiments were designed within the enhanced WRF-Chem modeling framework (Table 1).

1)  SET1: Historical Attribution Simulations (2013–2020).

The first set of simulations was conducted to identify the dominant drivers of $O_3$ variability during two major stages of CAAP, referred to as Phase I and Phase II. In total, 11 simulations were performed to isolate the effects of emission changes, meteorological variability, and aerosol-related mechanisms. To quantify the impact of anthropogenic emission changes alone, three simulations were conducted using fixed meteorological conditions from 2020, with all aerosol-related effects disabled (13E20M_NOALL, 17E20M_NOALL, and 20E20M_NOALL). Differences among these simulations represented the net $O_3$ response to emission evolution in the absence of aerosol feedbacks and meteorological variability. The contribution of meteorological variability was assessed through an additional set of simulations using fixed anthropogenic emissions from 2013 while varying meteorological conditions (2013, 2017, and 2020). Aerosol-related processes were excluded in these runs (13E13M_NOALL, 13E17M_NOALL, and 13E20M_NOALL), and the resulting differences quantified the meteorology-

driven component of O$_3$ changes. To evaluate aerosol effects (AEs), three parallel simulations were conducted for each emission year (2013, 2017, and 2020): (i) with all aerosol-related processes enabled (AEs), (ii) with heterogeneous chemistry disabled (NOHET), and (iii) with all aerosol effects turned off (NOALL). Pairwise comparisons among these simulations (e.g., AEs-NOHET, NOHET-NOALL, and AEs-NOALL) allowed the individual contributions of heterogeneous chemistry (HET), aerosol–radiation interactions (ARI), and their combined effects to be quantified. For example, the difference between 20E20M_AEs and 20E20M_NOHET isolated the HET contribution under 2020 emission conditions, whereas the comparison between 20E20M_NOHET and 20E20M_NOALL represented the ARI effect. This analytical framework was applied consistently across all emission years to characterize phase-resolved aerosol influences on O$_3$ trends. A schematic illustration of the experimental design and the associated O$_3$ responses was provided in Figure 1.

2)  SET2: Single-Precursor Sensitivity Experiments (2020 baseline).

The second group of simulations was designed to examine the nonlinear responses of O$_3$ to individual precursor emission controls under active aerosol effects. All experiments were based on the 2020 anthropogenic emissions inventory. For each simulation, emissions of one precursor (primary PM$_{2.5}$, NOx, volatile organic compounds (VOCs), SO$_2$, or NH$_3$) were reduced by 25% and 50%, while emissions of the remaining species were held constant. Reductions in primary PM$_{2.5}$ included both black carbon (BC) and organic carbon (OC).

3)  SET3: Multi-Pollutant Co-Reduction Experiments (Future Scenarios).

The third set of experiments explored potential O$_3$ responses under future emission mitigation pathways aligned with China's carbon peaking and carbon neutrality objectives. Coordinated reductions in all major anthropogenic emissions were applied, guided by the mid- and long-term projections reported by Cheng et al. (2021), who assessed China's air quality evolution under dual-carbon strategies. Their analysis suggested that anthropogenic emissions will decrease by approximately 26%–32% by 2030 relative to 2020 levels, followed by a slower reduction pace thereafter, reaching a maximum decline of about 31% by 2060 compared to 2030. Based on these projections, two representative reduction levels—30% and 50%—were selected to approximate emission conditions corresponding to the carbon peaking (2030) and carbon neutrality (2060) targets, respectively. To further characterize the nonlinear O$_3$ response under increasingly stringent mitigation, a series of additional co-control scenarios spanning 10%, 20%, 40%, 60%, 70%, 80%, and 90% reductions was implemented. Across all future experiments, emissions of primary PM$_{2.5}$, NOx, VOCs, SO$_2$, and NH$_3$ were scaled down proportionally, reflecting a coordinated multi-pollutant mitigation framework. Aerosol-related processes were consistently enabled in all simulations to preserve realistic aerosol–O$_3$ feedbacks.

**Table 1** Overview of WRF-Chem numerical experiments.

| Scenario sets | Scenario ID | Anthropogenic emissions | Meteorology | HET[a] | ARI[b] |
|---|---|---|---|---|---|

| | Scenario | | | HET[a] | ARI[b] |
|---|---|---|---|---|---|
| | 20E20M_AEs | | | √ | √ |
| | 20E20M_NOHET | 2020 | | × | √ |
| | 20E20M_NOALL | | | × | × |
| | 17E20M_AEs | | | √ | √ |
| | 17E20M_NOHET | 2017 | 2020 | × | √ |
| SET1 | 17E20M_NOALL | | | × | × |
| | 13E20M_AEs | | | √ | √ |
| | 13E20M_NOHET | 2013 | | × | √ |
| | 13E20M_NOALL | | | × | × |
| | 13E13M_NOALL | 2013 | 2013 | × | × |
| | 13E17M_NOALL | 2013 | 2017 | × | × |
| | CUT_PM₂.₅_25/50 | 25 (50) % reduction in $PM_{2.5}$ in 2020 | | | |
| | CUT_NOx_25/50 | 25 (50) % reduction in NOx in 2020 | | | |
| SET2 | CUT_VOCs_25/50 | 25 (50) % reduction in VOCs in 2020 | | | |
| | CUT_NH₃_25/50 | 25 (50) % reduction in $NH_3$ in 2020 | | | |
| | CUT_SO₂_25/50 | 25 (50) % reduction in $SO_2$ in 2020 | | | |
| | CUT_MEIC_10 | 10% reduction in 2020 | | | |
| | CUT_MEIC_20 | 20% reduction in 2020 | | | |
| | CUT_MEIC_30 | 30% reduction in 2020 | | | |
| | CUT_MEIC_40 | 40% reduction in 2020 | 2020 | √ | √ |
| SET3 | CUT_MEIC_50 | 50% reduction in 2020 | | | |
| | CUT_MEIC_60 | 60% reduction in 2020 | | | |
| | CUT_MEIC_70 | 70% reduction in 2020 | | | |
| | CUT_MEIC_80 | 80% reduction in 2020 | | | |
| | CUT_MEIC_90 | 90% reduction in 2020 | | | |

HET[a]: Heterogeneous chemistry (HET) was activated by setting the heterogeneous reaction switch to 1.
ARI[b]: Aerosol–radiation interaction (ARI) was activated by turning on the aerosol–radiation feedback (aer_ra_feedback = 1)
and by linking aerosol optical properties to the photolysis calculation.

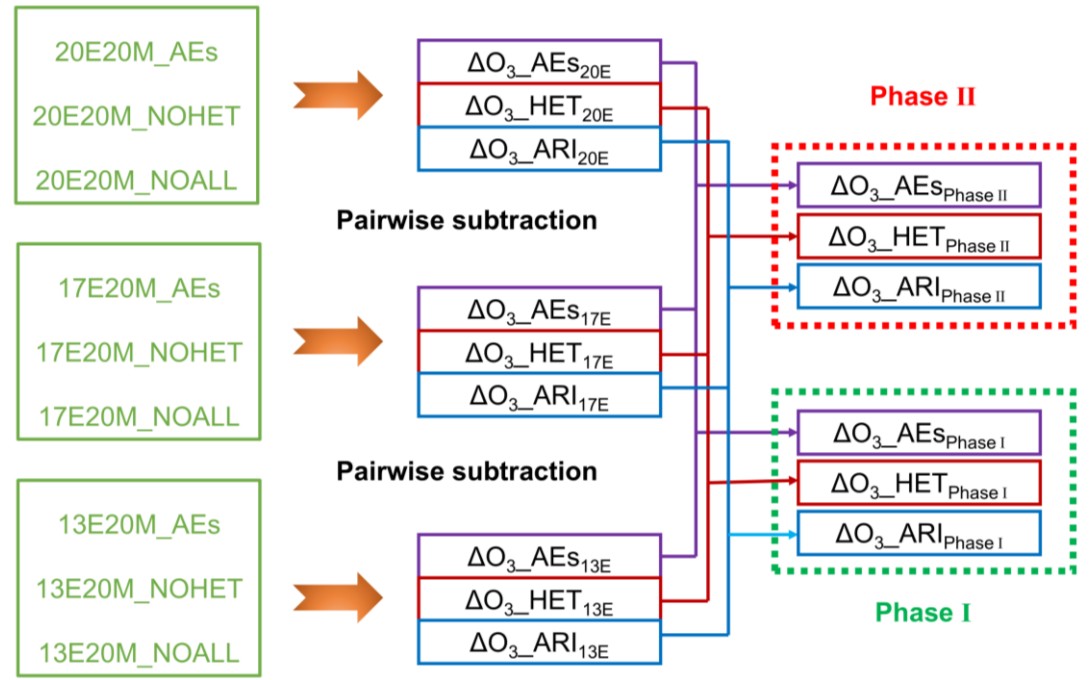


**Figure 1** Conceptual diagram illustrating the scenario design and the associated ozone responses to aerosol-mediated processes
during the CAAP phases. Note: HET=heterogeneous chemistry, ARI=aerosol-radiation interaction, AEs=aerosol effects
(HET+ARI). Scenario IDs such as "13E20M" refer to emission year 2013 with 2020 meteorology.

**2.4 Historical changes in emissions and observed $O_3$**

Interannual changes in six key species—$SO_2$, primary $PM_{2.5}$, BC, OC, NOx, and VOCs—at the provincial scale in the
YRD during 2013–2020 was presented in Figure S2. Over this period, emissions of most pollutants declined substantially,
with the exception of VOCs. Cumulatively, $SO_2$, primary $PM_{2.5}$, BC, OC, and NOx emissions were reduced by 69.7%, 46.9%,
40.4%, 38.0%, and 27.9%, respectively. During the first phase of the CAAP (Phase I), control strategies were predominantly
oriented toward particulate matter abatement. As a result, primary $PM_{2.5}$, BC, and OC emissions decreased markedly by 37.0%,
30.0%, and 27.3%, respectively. Concurrently, notable reductions were achieved for major gaseous precursors, with $SO_2$ and
NOx declining by 56.4% and 19.8%. In contrast, the absence of explicit VOCs-targeted measures during this stage led to a
7.1% increase in VOCs emissions (Li et al., 2019b). The second phase of the CAAP (Phase II) was characterized by a transition
toward more coordinated regulation of NOx and VOCs. Although emissions of $SO_2$, NOx, and $PM_{2.5}$ continued to decrease,
the overall pace of reduction was slower than that observed in Phase I. Specifically, NOx and VOCs emissions declined by
7.4% and 4.6%, respectively. Nevertheless, when considering the entire 2013–2020 period, VOCs emissions in the YRD still
exhibited a net increase of 2.2%. From a spatial perspective, emission reductions were most pronounced in the northwestern
and central subregions of the YRD (Figure S3), a pattern that aligns with national emission reduction trends and is consistent
with previous regional assessments (Liu et al., 2023a; Yan et al., 2024).
In addition to modifying emissions, the CAAP brought about substantial changes in observed $O_3$. Figure 2 illustrated the
annual variation of the MDA8 $O_3$ in winter and summer across the YRD based on ground-based observations from 2013 to
2024. In winter, $O_3$ increased by approximately 7 μg m$^{-3}$ during 2013–2017, at an average annual growth rate of 3%. This trend
reversed during 2017–2020, with a decrease of 4 μg m$^{-3}$ (2% per year), followed by a modest increase of 2.2 μg m$^{-3}$ (0.91%
per year) between 2020 and 2024. In summer, $O_3$ rose by 13.2 μg m$^{-3}$ during 2013–2017, continued to increase by 4.9 μg m$^{-3}$
from 2017 to 2020, and then declined sharply by 11.4 μg m$^{-3}$ during 2020–2024. These results suggested that in the early phase
of clean air efforts, the insufficient control of $O_3$ precursors contributed to significant increases in both winter and summer $O_3$.
However, stronger VOCs and NOx control measures in recent years appeared to mitigate this upward trend. A particularly
sharp drop in $O_3$ between 2020 and 2021 was likely caused by a combination of intensified emission reductions and unusual
meteorological conditions (Yin et al., 2021). Overall, observed MDA8 $O_3$ in the YRD increased by 12.1% in winter and 11.8%
in summer during 2013–2017. In the subsequent periods (2017–2020 and 2020–2024), winter $O_3$ levels first declined and then
rebounded, while summer $O_3$ initially rose and then decreased. The underlying causes of these contrasting patterns were
explored in detail in the Results section. Note that this study did not focus on the spatial distribution of $O_3$ changes, as this
topic has already been extensively examined in previous literature (Hu et al., 2025; Zhao et al., 2023).

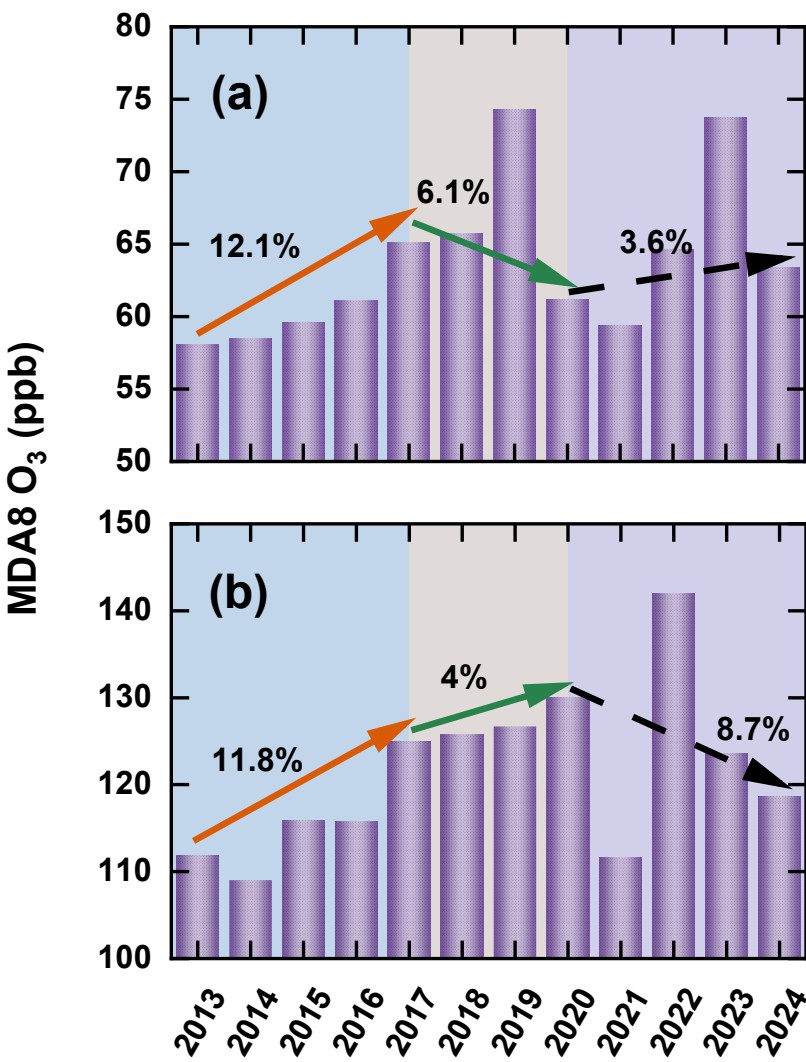

**Figure 2** Interannual variations in winter (a) and summer (b) MDA8 O$_3$ concentrations (ppb) across the YRD during 2013–2024, derived from continuous ground-based measurements.

## 3. Results and discussion

The accuracy of simulated meteorological parameters and pollutant concentrations under scenario (20E20M_AEs) has been thoroughly validated against ground-based observations in earlier work (Li et al., 2024a). As summarized in Table S2, the model reasonably captures the magnitude, seasonal variability of PM$_{2.5}$, O$_3$, as well as the major features of temperature, relative humidity, and wind speed. These results provide confidence in the model's ability to represent the atmospheric conditions relevant to the subsequent analysis.

### 3.1 Attribution of historical seasonal O$_3$ changes to emissions and meteorology

A set of attribution simulations (SET1) with aerosol processes disabled (NOALL) and fixed 2020 meteorology was conducted to isolate emission-driven O$_3$ variability in the YRD over the past decade, with the resulting responses shown in Figure 3. During Phase I, emission reductions unexpectedly led to O$_3$ increases of 6.3 ppb in winter and 1.3 ppb in summer. In contrast, Phase II witnessed coordinated NOx and VOCs controls, leading to O$_3$ reductions of 0.9 ppb (winter) and 1.5 ppb

(summer). These contrasting outcomes reflect the nonlinear chemistry of $O_3$ formation. While Phase I focused primarily on
reducing $PM_{2.5}$ and $SO_2$, VOCs emissions remained poorly regulated and even increased, enhancing photochemical activity.
In contrast, Phase II adopted a more balanced control strategy targeting both NOx and VOCs, which proved more effective in
mitigating $O_3$ pollution. Spatially, the strongest $O_3$ responses occurred in the northwestern and central parts of the YRD,
aligning with regions that experienced the largest emission reductions.
To assess the influence of meteorological conditions, we fixed anthropogenic emissions at 2013 levels and varied the
meteorological fields across years. Results revealed seasonally asymmetric impacts: meteorology contributed to wintertime
$O_3$ declines (1.7 ppb and 2.1 ppb during Phases I and II, respectively), but promoted summertime $O_3$ increases (1.4 ppb and
4.6 ppb). This highlighted a distinct seasonal asymmetry in meteorological influences on $O_3$. As summarized in Table S3,
changes in five key meteorological parameters (shortwave radiation (SW), temperature ($T_2$), relative humidity ($RH_2$), planetary
boundary layer height (PBLH), and wind speed ($WS_{10}$)) collectively explain these trends. In winter, lower radiation and $T_2$,
higher $RH_2$, and stronger $WS_{10}$ suppressed $O_3$ formation and accumulation. Conversely, summer conditions characterized by
higher radiation and $T_2$, coupled with lower $RH_2$ and weaker $WS_{10}$, favored $O_3$ build-up. Although this study does not explicitly
quantify the relative contributions of individual meteorological factors, prior studies (Liu et al., 2023a; Yan et al., 2024; Dai et
al., 2024) using multiple linear regression consistently identify SW and $T_2$ as dominant drivers. Figure S4 presented the spatial
distributions of meteorological changes during 2013-2020, revealing that the most pronounced shifts—especially in radiation
and temperature-occurred in the central YRD and were more significant in summer, consistent with stronger $O_3$ responses.
In summary, anthropogenic emission changes were the dominant drivers of winter $O_3$ increases during Phase I. These
findings are consistent with earlier research (Cao et al., 2022; Wu et al., 2022), which similarly highlighted that early-phase
air quality interventions-though effective in reducing $PM_{2.5}$-often overlooked the complex chemistry of $O_3$, particularly the
roles of VOCs and NOx, thereby unintentionally intensifying $O_3$ pollution. The transition to coordinated multi-pollutant control
strategies in Phase II enabled more effective $O_3$ mitigation. In addition, the role of meteorology was non-negligible. Our
findings, in line with those of Liu and Wang (2020), emphasize a pronounced seasonal asymmetry-meteorology suppressed
winter $O_3$ but enhanced summer levels. Notably, wintertime $O_3$ variability was primarily emission-driven during Phase I, but
increasingly influenced by meteorology in Phase II. In contrast, summer $O_3$ changes were consistently dominated by
meteorological variability across both phases. These insights underscore the need for future $O_3$ control strategies to account
for both emissions and meteorological variability, particularly in the context of climate change and evolving pollution regimes.
These externally driven $O_3$ changes provide the foundation for evaluating how aerosol-mediated processes further modulate
the emission-driven portion of the $O_3$ response.

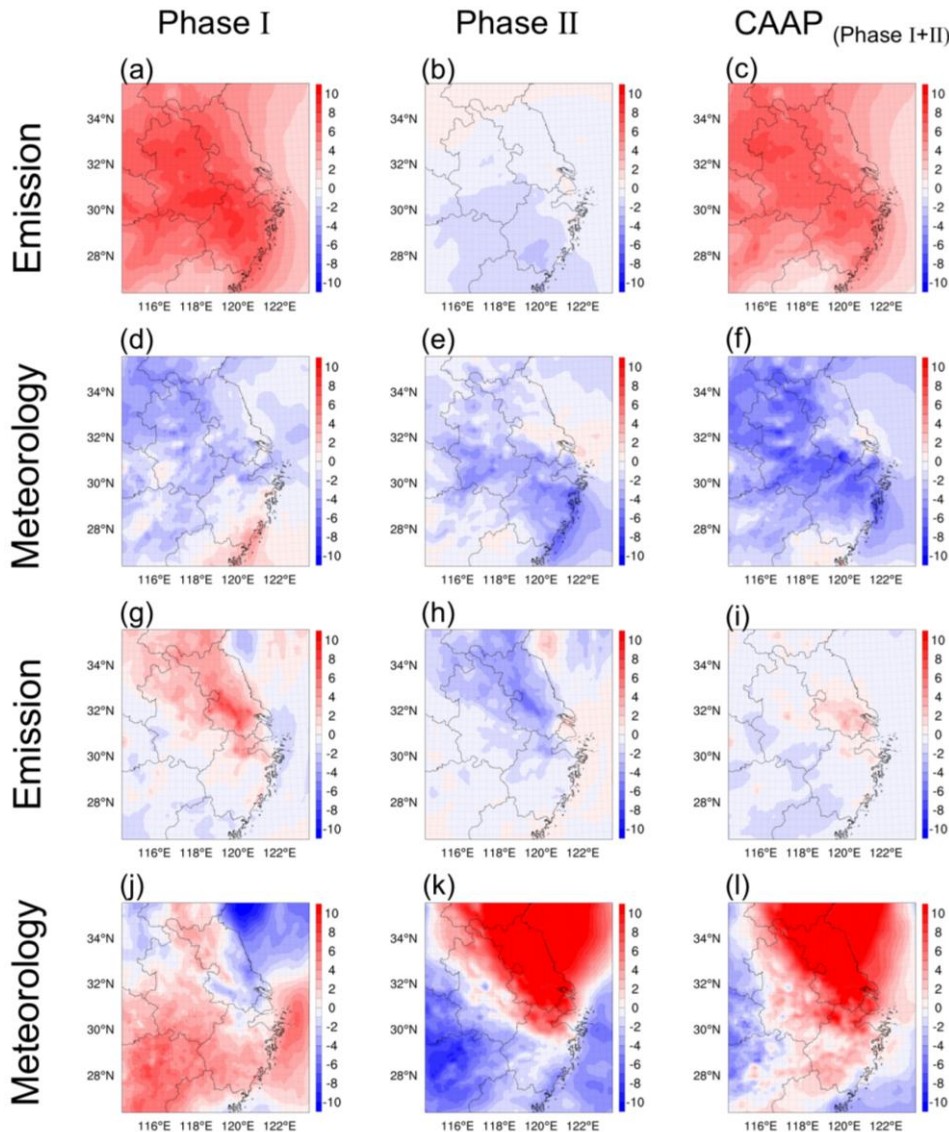


**Figure 3** Attribution of seasonal $O_3$ variations (ppb) in the Yangtze River Delta to emission reductions (a–c, g–i) and
meteorological influences (d–f, j–l) during Phase I and Phase II of the CAAP, with winter and summer results displayed in
the upper and lower rows, respectively.
**3.2  Aerosol multi-effects contributions to past seasonal $O_3$ variations**

266        Building on the external drivers identified in Section 3.1, we next examined how ARI and HET modified the emission-

reduction-driven $O_3$ response. Figure 4 illustrated the wintertime spatial patterns of $O_3$ changes driven by ARI and HET across
the YRD during both phases of the CAAP. In Phase I, ARI induced a significant $O_3$ increase of up to 1.14 ppb across the region,
while the contribution from HET was notably smaller at 0.32 ppb. This indicated that early aerosol reductions primarily
enhanced $O_3$ via increased solar radiation and associated meteorological feedbacks, rather than through the suppression of
radical uptake on particle surfaces. This finding contrasted with those of Li et al. (2019a), who—using GEOS-Chem
simulations—attributed $O_3$ increases over the BTH to reduced $HO_2$ uptake under declining $PM_{2.5}$. The discrepancy may stem
from differences in model representation; our framework explicitly incorporates both ARI-driven meteorological feedbacks
and the direct photolysis attenuation by aerosols, enabling a more comprehensive simulation of aerosol–radiation interaction.
During Phase II, the ARI-induced $O_3$ increase weakened to +0.74 ppb, and the contribution from HET became negligible or
slightly negative (−0.01 ppb). This suggested that ARI remained the dominant aerosol-related driver of winter $O_3$ variability,
while the influence of HET diminished. The reduced overall aerosol impact during this phase was consistent with smaller
primary $PM_{2.5}$ emission reductions (−8% in Phase II compared to −37% in Phase I). Summing the contributions from both
mechanisms, the total aerosol-driven $O_3$ enhancement reached +1.46 ppb in Phase I and +0.73 ppb in Phase II, culminating in
a net wintertime increase of +2.2 ppb over the CAAP period.

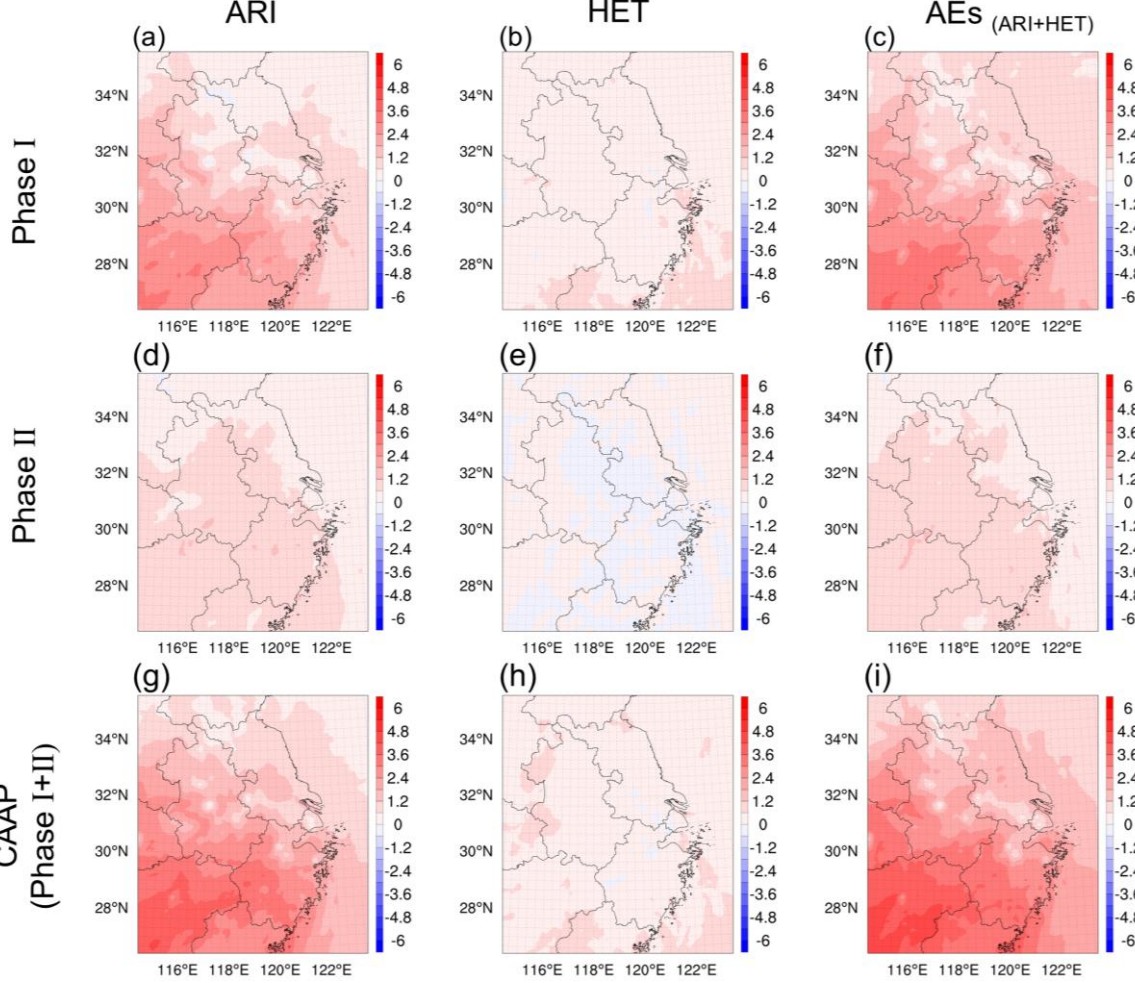


**Figure 4** Spatial distribution of winter $O_3$ changes (ppb) over YRD driven by ARI (a, d, g), HET (b, e, h) and their combined
effects (AEs, c, f, i) during two stages of the CAAP.

284         In contrast to winter, summertime $O_3$ responses to AEs revealed different dominant mechanisms and magnitudes, as

shown in Figure 5. In Phase I, HET played a more substantial role, contributing a 1.62 ppb increase, whereas ARI slightly
suppressed $O_3$ by 0.51 ppb. This pattern indicated that under high photochemical activity, reduced particulate matter
significantly weakened radical scavenging, thereby elevating $HO_2$ levels and promoting $O_3$ formation. During Phase II,
however, HET unexpectedly contributed a 2.86 ppb decreases in $O_3$, while ARI induced a 1.56 ppb enhancement. The HET-
driven decrease may be linked to complex nonlinear chemical responses under further reduced aerosol backgrounds, which
diminished the amplification effect of radical availability. Across both phases, HET consistently emerged as the primary driver
of summertime aerosol-related $O_3$ variability. When aggregated, aerosols contributed a 1.11 ppb increase in Phase I and a 1.30

ppb decrease in Phase II, yielding a modest net summer reduction of 0.19 ppb over the CAAP period.

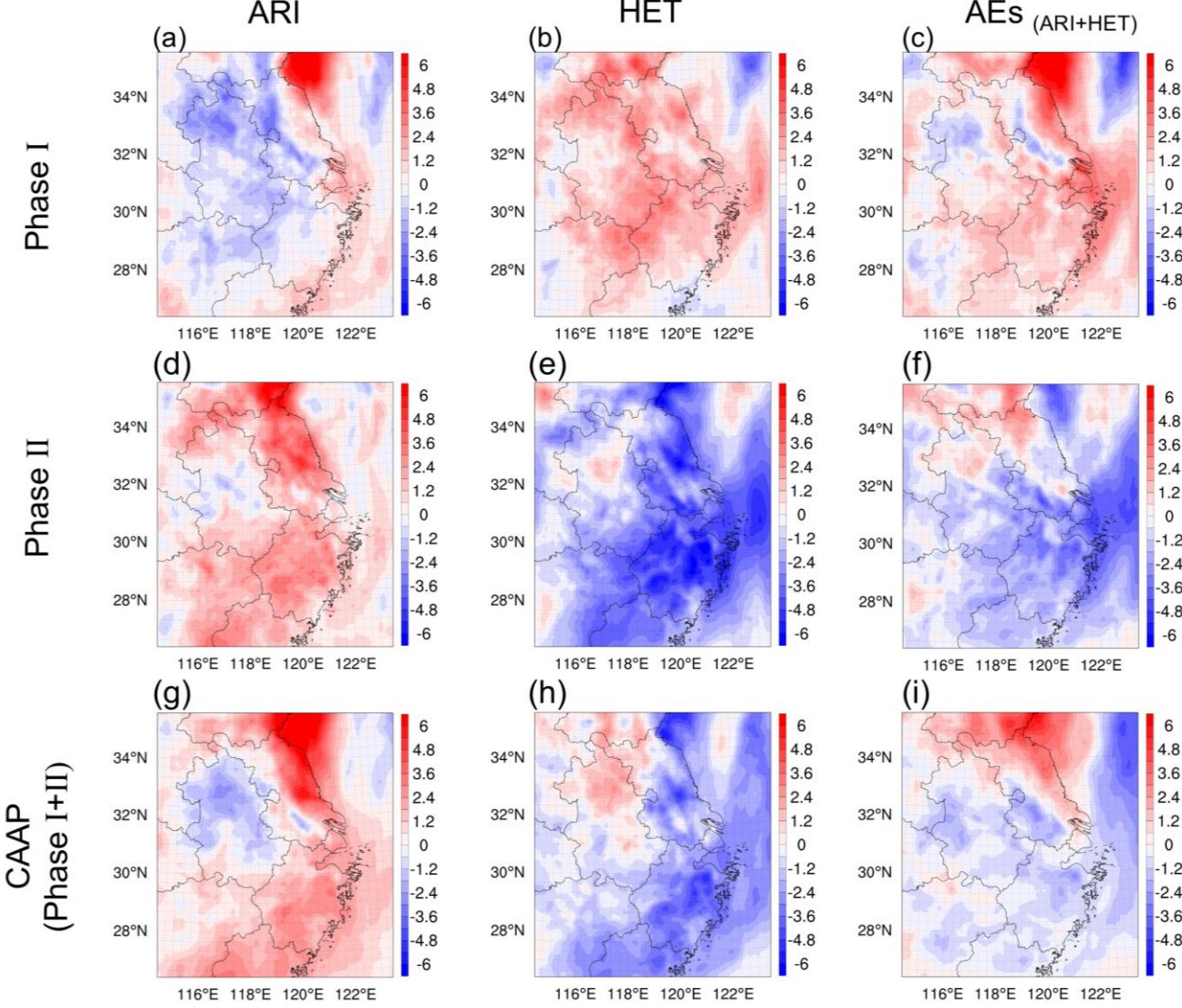

**Figure 5** Spatial distribution of summer $O_3$ changes (ppb) over YRD driven by ARI (a, d, g), HET (b, e, h) and their combined effects (AEs, c, f, i) during two stages of the CAAP.

To elucidate the underlying mechanisms of aerosol impacts on $O_3$, we examined the changes in key meteorological variables, photolysis rates, and $HO_2$ radical concentrations induced by ARI and HET during the two implementation phases of the CAAP. Figure 6 presented the variations in five key meteorological parameters and the $NO_2$ photolysis rate ($J_{NO2}$)) in winter and summer as influenced by ARI. The results indicated that ARI consistently enhanced $J_{NO2}$, SW, $T_2$, $WS_{10}$, and PBLH, while reducing $RH_2$ during winter across both phases. These modifications—especially increased SW and $T_2$—significantly facilitated photochemical $O_3$ production, thereby elevating $O_3$. Notably, the magnitude of these changes was substantially greater in Phase I than in Phase II, which can be attributed to the more pronounced reductions in aerosol emissions during the earlier phase. In summer, ARI and HET exerted contrasting influences on ground-level $O_3$, with their effects reversing between the two phases. ARI led to a slight decrease in $O_3$ (-0.51 ppb), likely due to enhanced vertical mixing from reduced aerosol extinction, which increased solar radiation and photolysis rates. However, the concurrent rise in temperature and PBLH may

have diluted surface $O_3$ in certain regions (Figure 6b), resulting in a net negative $O_3$ response to ARI during this phase. In
Phase II, the ARI-induced increases in $T_2$ and photolysis rates more effectively enhanced photochemical $O_3$ production.
Simultaneously, reductions in PBLH and $WS_{10}$ during this period suppressed vertical and horizontal $O_3$ dispersion (Figure 6b),
collectively leading to a net positive $O_3$ response (+1.56 ppb).

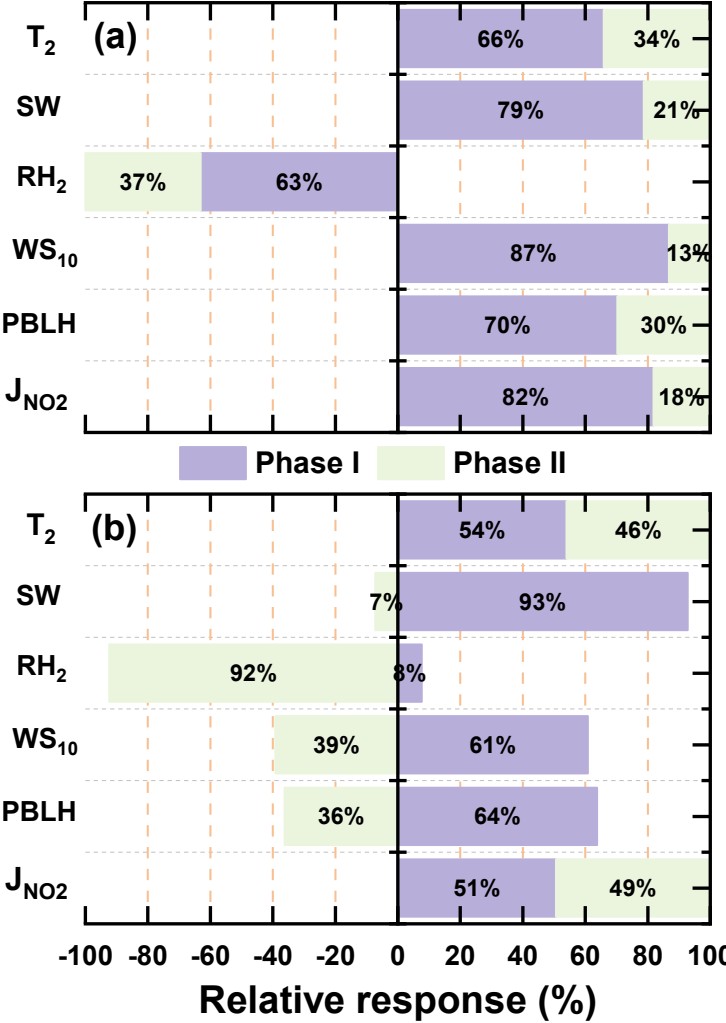

**Figure 6** ARI-driven relative responses of meteorological fields and photolysis rates in YRD during winter (a) and summer (b)
for the two CAAP stages.

During Phase I, the substantial reductions in aerosol mass and surface area primarily weakened $HO_2$ heterogeneous uptake,

as indicated by elevated $HO_2$ (Figure 7d). This reduction in radical loss increased the availability of $HO_2$ and OH, leading to

an enhancement in the photochemical ozone production term $P(O_3)$ (Dyson et al., 2023). In parallel, $N_2O_5$ also increased during

Phase I (Figure S5a), consistent with suppressed heterogeneous hydrolysis under reduced aerosol liquid water (ALW) and

diminished aerosol surface area (Brown and Stutz, 2012). The weakened $N_2O_5$ hydrolysis further limited nighttime conversion

of reactive nitrogen to $HNO_3$, maintaining NOx in more photochemically active forms (Ma et al., 2023b). Meanwhile,

heterogeneous $NO_2$ uptake—an important HONO source—was significantly reduced, consistent with the simulated decrease

in HONO (Figure S5d). The reduction in HONO slightly weakened early-morning radical initiation (Yu et al., 2022), but this

influence was outweighed by the strong enhancement in $HO_2$ and the limited conversion of NOx into $HNO_3$. As a result, HET

exerted a net positive contribution to $O_3$ (+1.62 ppb) in Phase I. In contrast, Phase II exhibited a fundamentally different chemical response. Although aerosol loadings continued to decrease, the relative importance of heterogeneous pathways shifted substantially. $HO_2$ declined during Phase II (Figure 7e), indicating a reduced radical pool and weaker propagation of daytime photochemical production. At the same time, $N_2O_5$ decreased markedly (Figure S5b), suggesting that nighttime $NO_3/N_2O_5$ chemistry became less effective at sustaining reactive nitrogen cycling under even lower aerosol surface area and ALW. Rather than promoting efficient nighttime NOx recycling, this suppression favored a net loss of reactive nitrogen through terminal sinks (e.g., $HNO_3$), shifting NOx partitioning toward less photochemically active forms and weakening daytime $P(O_3)$. Conversely, HONO concentrations rebounded during Phase II (Figure S5e). This increase reflects the altered balance between $NO_2$ uptake and nighttime NOx partitioning under reduced $N_2O_5$ hydrolysis. However, despite this HONO increase, its positive effect on radical initiation could not compensate for the combined decline in $HO_2$, weakened $N_2O_5$ hydrolysis, and enhanced $HNO_3$ formation (George et al., 2015). The joint effect was a net reduction in the morning radical pool and diminished photochemical $O_3$ production (-2.86 ppb). This multi-pathway adjustment explains the observed sign reversal of HET's effect on $O_3$ between the two phases and underscores the importance of considering the full suite of heterogeneous processes—rather than radical uptake alone—when interpreting aerosol-mediated $O_3$ responses. In future work, we plan to apply integrated process rate (IPR) diagnostics to more directly evaluate how individual heterogeneous pathways—such as $HO_2$ uptake, HONO formation, and $N_2O_5$ hydrolysis—shape the resulting $O_3$ responses. Coupled with continued improvements in heterogeneous chemistry parameterizations and more comprehensive constraints on radical, reactive nitrogen, and aerosol liquid water fields, this will enable a more detailed and process-resolved understanding of phase-dependent $O_3$ changes.

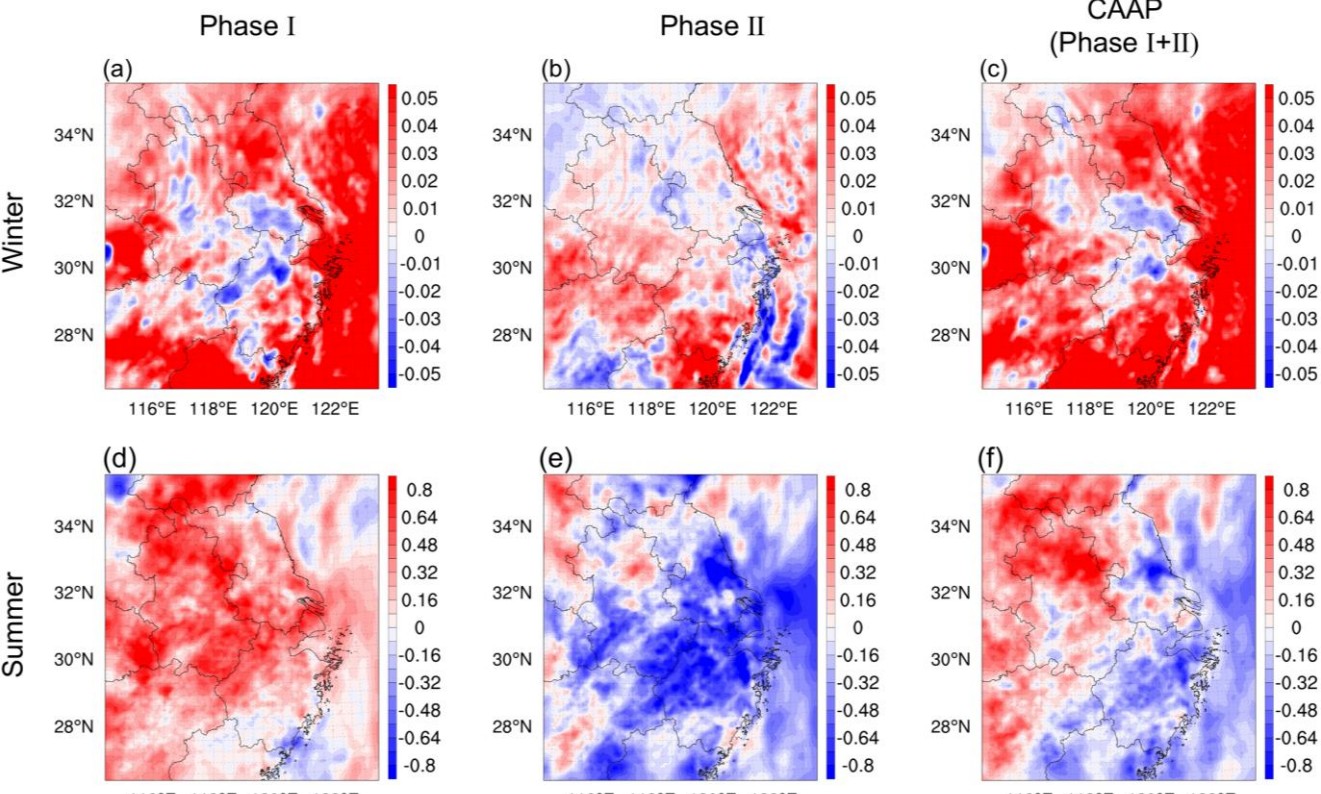

Figure 7 Spatial distributions of HO₂ concentration (ppt) changes induced by HET in winter (a-c) and summer (d-f) during two phases of the CAAP in YRD.

To further evaluate whether daytime and nighttime $O_3$ responses compensate within the daily mean metric, we examined the diurnal cycles of baseline $O_3$ concentration and the aerosol-mediated impacts (HET, ARI, and AEs) during Phase I, Phase II, and the overall CAAP period for both winter (Figure S6) and summer (Figure S7). Across all phases and both seasons, the dominant $O_3$ perturbations occur during daytime hours, coinciding with the photochemical peak at 14–16 LT. In winter, Phase I exhibits a pronounced daytime enhancement driven by ARI (up to ~2.41 ppb), whereas HET induces a consistently positive but comparatively weaker increase (up to ~0.49 ppb). In Phase II, the ARI-induced enhancement weakens notably (peaking at ~1.24 ppb), and HET-induced changes remain minor. In summer, the diurnal behavior more clearly reflects a daytime-dominated response. During Phase I, HET produces a marked midday $O_3$ enhancement (up to ~2.01 ppb), while ARI imposes a weaker yet persistent negative contribution. In contrast, Phase II is characterized by a strong HET-driven daytime $O_3$ decrease (maximum ~3.43 ppb), overwhelming the comparatively modest positive ARI effect. For all cases, nighttime $O_3$ changes share the same direction as daytime responses but remain substantially smaller in magnitude, insufficient to offset the daytime signals dominated by photochemistry. These diurnal patterns confirm that the phase-dependent $O_3$ responses to aerosol effects are not artifacts of day–night compensation in daily mean metrics, but instead arise from robust, daytime-dominant photochemical adjustments.

Previous studies showed that ARI and HET were not fully independent and could interact through aerosol–meteorology–chemistry feedbacks (Chen et al., 2019; Liu et al., 2023b; Kong et al., 2018; Li et al., 2020a). ARI-induced increases in near-surface relative humidity typically enhanced aerosol hygroscopic growth and expanded aerosol surface area. The resulting

increase in aerosol liquid water promoted gas-to-particle partitioning and facilitated aqueous- and surface-phase reactions,
thereby accelerating heterogeneous oxidation pathways involving $SO_2$ and NOx. The strengthened heterogeneous formation
of secondary inorganic aerosols further modified solar radiation and potentially intensified the ARI effect. In the present study,
our primary focus was to quantify the separate and combined contributions of ARI and HET to $O_3$ changes across different
stages of the CAAP. Accordingly, we isolated their individual impacts rather than examining their nonlinear coupling. We
acknowledged that ARI–HET interactions might also affect $O_3$ under certain chemical and meteorological conditions, and we
indicated that future work would incorporate dedicated coupled-sensitivity experiments to more explicitly quantify these
nonlinearities and their implications for $O_3$ formation.
Figure S8 illustrated the hierarchical relationships among the four factors analyzed in this section. Emission reductions
and meteorological variability constituted the external drivers of $O_3$ changes, whereas ARI and HET acted as aerosol-mediated
modulators that adjust the emission-reduction-driven $O_3$ responses. This framework motivated our presentation sequence,
where external drivers were examined first, followed by the modulation effects of ARI and HET. Figure 8 presented the relative
contributions of major driving factors to surface $O_3$ changes during the two phases of the CAAP. In winter, anthropogenic
emissions emerged as the dominant driver of $O_3$ increases during Phase I, contributing 6.3 ppb, primarily due to enhanced
photochemical production under VOCs-limited conditions. In contrast, Phase II saw a modest $O_3$ decline (0.9 ppb) resulting
from co-reductions in NOx and VOCs, suggesting improved control effectiveness through coordinated precursor mitigation.
Meteorological changes consistently exerted a suppressive effect on wintertime $O_3$, contributing −1.7 ppb and −2.1 ppb in
Phases I and II, respectively. AEs—mediated by ARI and HET—also contributed to $O_3$ accumulation, particularly in Phase I
(+1.46 ppb), though their influence weakened in Phase II (+0.73 ppb) due to the smaller reductions in aerosol loading. Overall,
the wintertime $O_3$ increase in Phase I was jointly driven by emissions and aerosol-related processes, while the slight decline
in Phase II reflected the synergistic benefits of emission reductions and favorable meteorological conditions. In contrast, the
attribution profile for summer revealed a dominant role of meteorology. Meteorological variability accounted for a substantial
$O_3$ increase in Phase II (+4.6 ppb), outweighing the contributions of emission changes. The effect of emission reductions on
summer $O_3$ was limited and nonlinear: a slight increase (+1.3 ppb) was observed in Phase I, followed by a minor decline (−1.5
ppb) in Phase II, indicative of a photochemical regime with weak emission sensitivity. Aerosol-related effects exhibited strong
seasonal contrasts. HET was the dominant mechanism influencing $O_3$ in both summer phases, albeit with opposite signs—
enhancing $O_3$ by 1.62 ppb in Phase I but reducing it by 2.86 ppb in Phase II. These contrasting effects likely reflect differences
in $HO_2$ uptake efficiency under evolving humidity and temperature conditions. ARI effects were comparatively modest, leading
to a slight $O_3$ decrease in Phase I (0.51 ppb) and an increase in Phase II (1.56 ppb), likely driven by enhanced photolysis and
reduced vertical mixing.
Collectively, these results highlight the evolving interplay among emission control efforts, meteorological conditions, and

aerosol effects in shaping surface $O_3$ trends. While anthropogenic emissions primarily drove winter $O_3$ increases during the early phase of the CAAP, the roles of meteorology and aerosol processes became increasingly prominent in summer and in the later policy phase. This multi-factor attribution framework aligns well with prior modeling and observational studies in eastern China (Zhu et al., 2021; Zhou et al., 2019). For example, Liu et al. (2023a) demonstrated that declining $PM_{2.5}$ levels enhanced $O_3$ formation by weakening $HO_2$ radical scavenging, particularly under VOCs-limited regimes—a conclusion consistent with our wintertime results. Similarly, Yang et al. (2019) highlighted the growing influence of meteorological variability in recent years as the sensitivity of $O_3$ to emission changes has diminished. Our study extends this knowledge base by providing phase-resolved attribution and explicitly separating the effects of ARI and HET. Notably, the reversal of HET-driven $O_3$ responses in summer—from enhancement to suppression—has rarely been quantified and underscores the importance of dynamically characterizing aerosol–ozone interactions under evolving atmospheric and policy contexts.

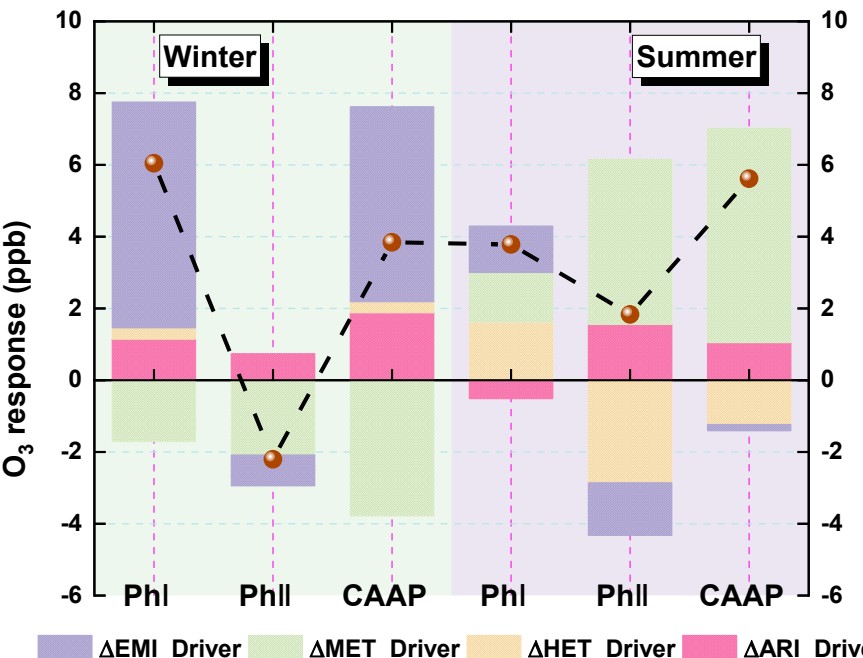

**Figure 8** Quantitative attribution of surface $O_3$ changes over the Yangtze River Delta (YRD) during the two CAAP phases. Contributions from emission reductions (EMI), meteorological variability (MET), aerosol–radiation interactions (ARI), and heterogeneous chemistry (HET) are shown for winter (left) and summer (right).

**3.3  $O_3$ responses to precursor emission reductions under aerosol effects**

Before presenting the simulation results, we first assessed the $O_3$ chemical regimes over YRD using the widely adopted $H_2O_2/HNO_3$ ratio (Jeon et al., 2018; Peng et al., 2011; Hammer et al., 2002; Zhang et al., 2000). This metric serves as a diagnostic indicator of $O_3$ production sensitivity, with ratios <0.6 indicating VOCs-limited conditions, >0.8 denoting NOx-limited regimes, and intermediate values representing transitional states. Figure S9 showed the spatial distribution of this ratio under the baseline scenario (20E20M_AEs). The analysis reveals that wintertime $O_3$ formation is predominantly VOCs-limited across the YRD, while in summer, most areas exhibit transitional or NOx-limited regimes, except parts of Anhui Province. Figure 9 displayed the simulated $O_3$ responses to precursor reductions in both seasons. The results highlight strong seasonal

differences and nonlinear sensitivities depending on chemical regime. In winter, reductions in primary $PM_{2.5}$ and NOx led to
substantial $O_3$ increases. Specifically, 25% and 50% reductions in $PM_{2.5}$ increased $O_3$ by 0.7 ppb and 1.5 ppb, respectively,
while NOx reductions caused even larger enhancements of 4.8 ppb and 10.2 ppb. These increases primarily stem from
weakened aerosol suppression mechanisms—namely reduced heterogeneous uptake and increased photolysis rates—which
enhance radical availability and photochemical activity. Additionally, under VOCs-limited conditions, NOx reductions
diminish $O_3$ titration by NO, further contributing to $O_3$ accumulation. Among all precursors, NOx reductions produced the
most pronounced $O_3$ increase. In contrast, $NH_3$ and $SO_2$ reductions exerted negligible impacts on $O_3$, underscoring their limited
roles in direct $O_3$ photochemistry. VOCs controls, on the other hand, effectively suppressed $O_3$ formation, with 25% and 50%
reductions yielding decreases of 2.7 ppb and 5.6 ppb, respectively. In summer, $O_3$ responses followed broadly similar trends
but with different magnitudes. Reducing $PM_{2.5}$ and NOx increased $O_3$ by 2 ppb and 4.3 ppb ($PM_{2.5}$) and 0.8 ppb and 1.6 ppb
(NOx), respectively. Notably, the $O_3$ increase associated with $PM_{2.5}$ reductions exceeded that from NOx cuts, underscoring the
critical role of particulate matter in regulating radical chemistry via aerosol-mediated pathways. VOCs reductions remained
the only control strategy that consistently decreased $O_3$, lowering concentrations by 1.6 ppb and 3.4 ppb for 25% and 50%
reductions, respectively. Again, $NH_3$ and $SO_2$ reductions had negligible effects. Collectively, these findings suggest that
continued $PM_{2.5}$-targeted controls may inadvertently worsen $O_3$ pollution under active AEs, particularly in summer. In contrast,
VOCs mitigation remains the most robust and seasonally effective strategy for $O_3$ reduction.

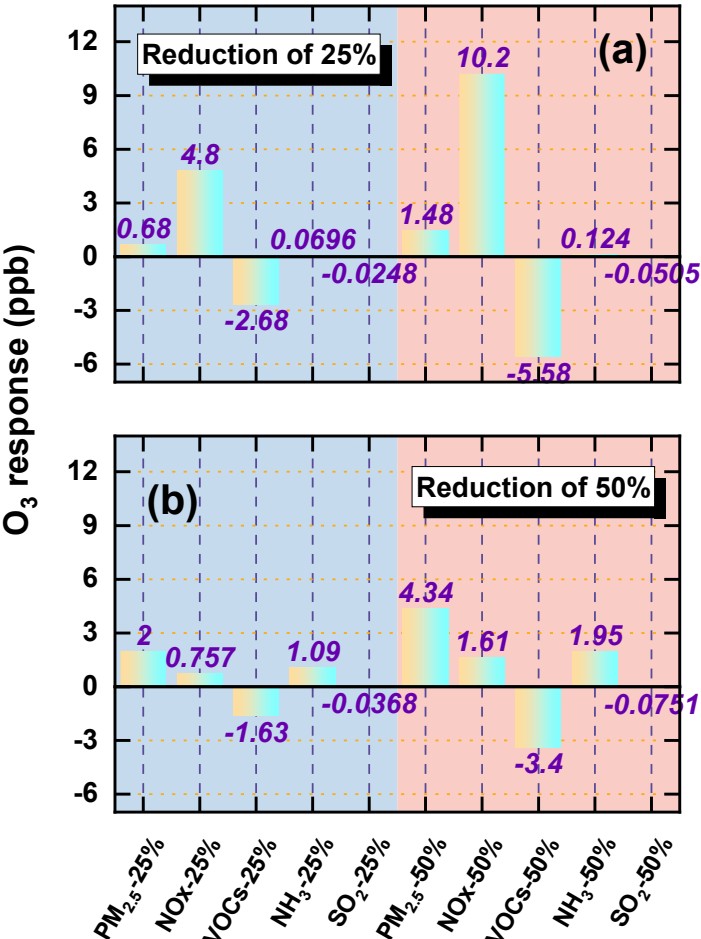

**Figure 9** $O_3$ concentration changes (ppb) in response to 25% and 50% reductions in precursor emissions over YRD during winter (a) and summer (b).

Figure S10 presented the distribution of $O_3$ changes under 25% and 50% precursor reductions for both seasons. Strong seasonal contrasts and regional gradients in $O_3$ responses are evident. Reductions in $PM_{2.5}$ consistently caused widespread $O_3$ increases across the YRD, with the most pronounced enhancements in northwestern inland regions—particularly southern Jiangsu and central-to-northern Anhui—where historically high aerosol burdens make $O_3$ formation especially sensitive to weakened aerosol suppression (via ARI and HET). Conversely, coastal cities such as Shanghai and eastern Zhejiang exhibited smaller $O_3$ increases, reflecting their lower baseline aerosol concentrations and weaker aerosol feedbacks. VOCs reductions led to the largest $O_3$ decreases in urban corridors, particularly along the Shanghai–Nanjing–Hangzhou (SNH) axis, where VOCs emissions are elevated and $O_3$ formation is strongly VOCs-sensitive. NOx reductions yielded seasonally opposite effects: in winter, $O_3$ increased broadly across the YRD, while in summer, decreases were observed in most regions except Anhui Province. These patterns align with seasonal chemical regimes inferred from $H_2O_2/HNO_3$ ratios—VOCs-limited in winter and NOx-limited or transitional in summer. $NH_3$ and $SO_2$ reductions produced negligible spatial effects in both seasons, reinforcing their limited involvement in direct $O_3$ photochemistry. These spatially heterogeneous responses highlight the need for geographically differentiated control strategies. Regions with historically high aerosol pollution are more likely to experience unintended $O_3$ increases following $PM_{2.5}$ or NOx reductions. Conversely, VOCs control provides consistent and widespread

$O_3$ benefits across both seasons, making it a key lever for achieving co-benefits in both $PM_{2.5}$ and $O_3$ mitigation.
To better understand the temporal dynamics of $O_3$ responses, we analyzed diurnal variations in four representative cities—
Shanghai, Nanjing, Hangzhou, and Hefei—under 50% reductions of individual precursors (Figure S11). In winter, NOx
reductions led to substantial $O_3$ increases during afternoon hours (14:00–17:00), particularly in urban centers like Shanghai
and Hangzhou, where enhancements exceeded 15 ppb. These increases reflect the dual effect of diminished NO titration and
enhanced photochemical activity. $PM_{2.5}$ reductions also caused moderate $O_3$ increases from late morning to early afternoon,
underscoring the influence of both ARI and HET. VOCs reductions induced midday $O_3$ declines (12:00–15:00) exceeding 5
ppb, consistent with VOCs-limited wintertime chemistry. In summer (Figure S12), VOCs reductions suppressed $O_3$ throughout
the daytime, with maximum declines reaching up to 25 ppb in early afternoon, reaffirming the effectiveness of VOCs control.
In contrast, $PM_{2.5}$ reductions led to notable $O_3$ increases during photochemically active hours (11:00–16:00), highlighting the
critical role of aerosols in modulating radical cycles and $O_3$ production. Overall, these diurnal profiles underscore the time-
sensitive nature of $O_3$ responses to precursor emission reductions. They emphasize the necessity for temporally and spatially
refined control strategies that account for local photochemical regimes, emission structures, and AEs.
**3.4   Future $O_3$ responses to Carbon neutrality–driven emission reductions considering aerosol effects**
We performed a suite of sensitivity experiments using the 2020 anthropogenic emissions as the baseline to examine
prospective $O_3$ responses to emission mitigation under China's carbon peaking and carbon neutrality pathways. As shown in
Figure 10, $O_3$ exhibited pronounced seasonal variability in response to progressive emission reductions. In winter, regional
mean $O_3$ increased monotonically with the magnitude of emission cuts, rising from +2.1% under the 10% reduction scenario
to +14.6% under the 90% scenario. This counterintuitive increase is primarily attributed to two synergistic mechanisms: (1)
reduced $O_3$ titration resulting from NOx emission reductions, and (2) weakened aerosol-mediated $O_3$ suppression due to lower
aerosol loads, which diminish both ARI and HET processes. The reduced availability of aerosol surfaces and optical attenuation
enhances photolysis rates and radical propagation, thereby promoting $O_3$ accumulation. In contrast, summer $O_3$ declined
steadily with increasing emission reductions, from -1.5% to -16.5% across the same range. This decline reflects the dominance
of VOCs-limited or transitional photochemical regimes in the region during summer, where coordinated reductions in NOx
and VOCs effectively suppress $O_3$ formation. These results underscore the seasonal asymmetry of $O_3$ responses under the
carbon-neutrality-aligned emission trajectories used in this study—namely the proportional precursor-reduction pathways
designed to reflect long-term, economy-wide emission declines. While such stringent reductions may inadvertently aggravate
wintertime $O_3$ pollution, they yield substantial co-benefits for summer $O_3$ mitigation. The spatial distribution of $O_3$ changes
under these scenarios, presented in Figure S13, further corroborates the contrasting seasonal patterns. In winter, $O_3$ increases
were most pronounced in inland areas of northern Anhui and central Jiangsu—regions characterized by historically high
aerosol burdens and stronger aerosol-mediated $O_3$ suppression. As emissions decline, the weakening of both aerosol effects
and NOx titration leads to a disproportionate $O_3$ rebound in these locations. The largest summer $O_3$ reductions observed in
densely populated urban corridors such as Shanghai, Nanjing, and Hangzhou. These metropolitan areas, with high precursor
emissions and transitional or NOx-limited chemical regimes, are particularly responsive to coordinated VOCs and NOx
controls. The spatial heterogeneity in $O_3$ responses highlights the necessity of designing region-specific and seasonally
adaptive emission control strategies. Differentiated approaches are essential given the diverse pollution histories, chemical
sensitivities, and aerosol–ozone coupling characteristics across the YRD.

483        Overall, these findings suggest that carbon neutrality–driven emission pathways, if carefully managed, can yield

significant summertime $O_3$ mitigation benefits, but must be complemented with targeted wintertime strategies to avoid adverse
trade-offs. The proportional 10-90% reductions applied uniformly across all pollutant species were designed as an idealized
framework to systematically examine nonlinear $O_3$ responses under consistent boundary conditions. In practice, however,
future emission pathways are expected to exhibit pronounced sectoral and spatial heterogeneity—for example, $SO_2$ and
primary $PM_{2.5}$ typically decline faster than VOCs and $NH_3$, and the pace of reductions varies across industrial, transportation,
and residential sectors. Such differences may influence the magnitude of $O_3$ responses and the balance among precursor
contributions. Recognizing this limitation, future work will incorporate sector-resolved and scenario-specific emission
pathways to provide a more realistic assessment of $O_3$ sensitivity under evolving emission structures.

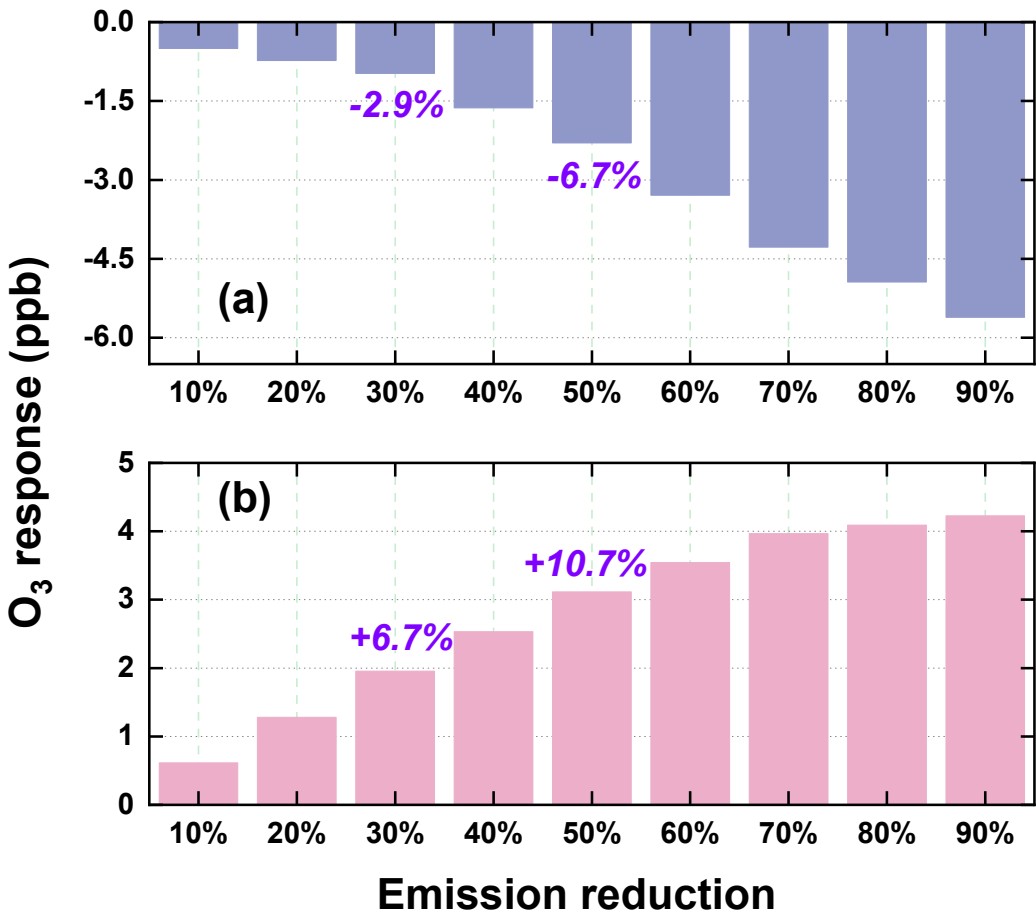


**Figure 10** Seasonal variations in $O_3$ concentrations (ppb) projected under a range of emission reduction intensities (10%–90%),

including representative scenarios for carbon peaking (30%) and carbon neutrality (50%), referenced to 2020 conditions with

aerosol-related processes accounted for. Results for summer and winter are displayed in the upper and lower panels,

respectively.

**3.5 Discussion and policy implications**

This study presented a comprehensive assessment of $O_3$ responses to emission reductions under both the CAAP and future carbon neutrality pathways, explicitly considering aerosol effects. Our findings underscore that while emission control measures have been effective in substantially lowering $PM_{2.5}$, they may yield unintended consequences for $O_3$ pollution—particularly under VOCs-limited regimes during winter. Specifically, aerosol-induced enhancements in $O_3$—via weakened heterogeneous chemistry (HET) and increased photolysis (ARI)—highlight the necessity of accounting for multiphase feedback mechanisms in the design of future air quality strategies. Our phase-resolved, seasonally differentiated attribution analysis suggests that coordinated reductions in VOCs and NOx are critical for effective $O_3$ mitigation, especially in summer when photochemical activity is most intense. Furthermore, the spatial heterogeneity of $O_3$ responses calls for region-specific strategies. For instance, in inland areas with historically high aerosol burdens, the potential for $O_3$ rebound due to weakened aerosol suppression is more pronounced, necessitating tailored mitigation approaches. In contrast, urban corridors such as the Shanghai–Nanjing–Hangzhou (SNH) axis—characterized by high VOCs emissions and transitional or NOx-limited regimes—stand to benefit most from targeted VOCs controls, particularly under future carbon-neutrality-driven reductions.

Uncertainties in HET parameterizations also introduce potential variability into the estimated $O_3$ responses. The uptake coefficients ($\gamma$) for $HO_2$, $NO_2$, and $N_2O_5$ depend on aerosol liquid water content, acidity, ionic strength, and particle composition (Jacob, 2000), yet these dependencies remain imperfectly constrained in current atmospheric models. As a result, uncertainties in these parameters may alter the magnitude of individual heterogeneous pathways simulated in this study. For example, higher assumed $HO_2$ uptake would strengthen radical loss and could reduce the positive HET contribution during Phase I, whereas larger $N_2O_5$ hydrolysis rates would enhance nighttime conversion of NOx to $HNO_3$ and potentially intensify the negative HET influence in Phase II. Likewise, uncertainties in $NO_2$ uptake and HONO yields could modulate early-morning radical initiation and shift the balance between radical propagation and reactive nitrogen recycling. Importantly, while such uncertainties may influence the absolute magnitude of HET-induced $O_3$ perturbations, they are unlikely to overturn the direction of the response. Prior modeling studies provide support for this robustness. For instance, Shao et al. (2021) showed that varying $\gamma_{HO2}$ between 0.2 and 0.08 altered the magnitude of the $O_3$ increase driven by reduced $HO_2$ heterogeneous uptake—from approximately 6% (consistent with the ~7% reported by Li et al. (2019a) to about 2.5% during 2013–2016—yet the effect remained positive in all cases. These findings indicate that although heterogeneous uptake assumptions can change the amplitude of the response, the sign of the $O_3$ change is preserved because the underlying chemical mechanism (reduced radical loss leading to enhanced photochemical production) remains the same. By analogy, the phase-dependent sign reversal identified in our study reflects a structural shift in the competition among $HO_2$ uptake, $N_2O_5$ hydrolysis, and HONO formation pathways, and is therefore unlikely to be reversed by plausible uncertainties in individual uptake coefficients. Our future studies will incorporate dedicated sensitivity simulations and integrated process rate (IPR) diagnostics to more systematically quantify how uncertainties in

heterogeneous chemistry parameterizations propagate into $O_3$ simulations. Improvements in observational constraints on aerosol acidity, liquid water content, and heterogeneous reaction rates will further strengthen mechanistic understanding and reduce uncertainty in model-based assessments of aerosol–$O_3$ interactions under evolving emission pathways.

It is worth emphasizing that all simulations were performed under a fixed-meteorology configuration, which was designed to isolate the influences of aerosol processes and emission changes on $O_3$ by suppressing interannual meteorological variability. This strategy improves the interpretability of attribution results by reducing confounding weather effects, but it inevitably constrains the model's ability to capture $O_3$ variability associated with meteorological extremes, such as heat waves or anomalous circulation patterns. As a result, caution is warranted when extending these findings to long-term evolutions or climate-change contexts, where interactions between emissions and meteorology may substantially alter $O_3$ responses. Future work will explicitly address this limitation by conducting additional sensitivity experiments with time-varying meteorological conditions.

These findings carry timely relevance for China's national climate and environmental goals. As outlined in the 14th Five-Year Plan for Ecological and Environmental Protection and the 2060 Carbon Neutrality Roadmap, deep multi-sector emission cuts are pivotal for achieving synergistic benefits between air quality improvement and climate change mitigation. Our results demonstrate that under prevailing atmospheric chemical regimes—especially during winter—aggressive reductions in primary $PM_{2.5}$ and NOx may inadvertently exacerbate $O_3$ pollution unless accompanied by VOCs-focused controls and regionally tailored strategies. In light of these findings, we advocate for an integrated policy framework that (i) coordinates VOCs and NOx reductions according to regional $O_3$ sensitivity, (ii) strengthens VOCs monitoring and inventory resolution at the city level, and (iii) explicitly incorporates aerosol effects in both short-term air pollution forecasting and long-term carbon-neutrality scenarios. Such targeted and mechanism-informed strategies will help bridge the current policy gap between $PM_{2.5}$ control and $O_3$ pollution mitigation, while ensuring co-benefits under evolving climate objectives.

**4.    Conclusions**

We employed a phase- and season-specific WRF-Chem framework that explicitly accounted for aerosol–radiation interactions and heterogeneous chemistry to characterize aerosol-driven modulation of $O_3$ over the YRD from 2013 to 2024. Through combined analyses of emission transitions, meteorological variability, and carbon-neutrality–oriented scenarios, this study provides an integrated assessment of the mechanisms governing historical $O_3$ changes and future responses to precursor emission controls.

$O_3$ exhibited a distinct rise–fall trajectory over the past decade, shaped by complex interactions among emission reductions, meteorological changes, and aerosol effects. During Phase I, substantial reductions in $PM_{2.5}$ and $SO_2$, coupled with inadequate VOCs controls, led to significant wintertime $O_3$ increases (6.29 ppb) and modest summer increases (1.28 ppb). In

Phase II, more balanced reductions in NOx and VOCs effectively suppressed $O_3$ formation. Meteorological variability also
exhibited seasonally asymmetric impacts—suppressing $O_3$ in winter but enhancing accumulation in summer. While wintertime
$O_3$ changes were primarily driven by emissions, summertime variations were dominated by meteorological factors. Aerosol
effects further modulated $O_3$ concentrations through seasonally distinct mechanisms. In winter, ARI played the dominant role:
the substantial aerosol reductions in Phase I enhanced solar radiation and boundary layer development, promoting $O_3$ formation
(1.14 ppb); these effects weakened in Phase II (0.73 ppb).   Summer $O_3$ was more sensitive to HET. In Phase I, aerosol
decreases weakened heterogeneous radical uptake, enhancing $O_3$ formation (+1.62 ppb). In Phase II, however, the net HET
effect reversed sign (–2.86 ppb), driven by shifts in multiple heterogeneous pathways—including changes in radical uptake,
HONO and $N_2O_5$ chemistry, and aerosol liquid water—rather than radical scavenging alone.
Accounting for aerosol effects, precursor emission reductions elicited marked seasonal and spatial $O_3$ responses. In winter,
a 50% reduction in VOCs effectively suppressed $O_3$ by 5.58 ppb, whereas equivalent reductions in NOx and $PM_{2.5}$ increased
$O_3$ by 10.2 ppb and 1.48 ppb, respectively—primarily due to weakened $O_3$ titration and radical loss processes. In summer,
reductions in $PM_{2.5}$ led to greater increases in $O_3$ than NOx (4.34 ppb vs. 1.61 ppb under the 50% reduction scenario),
highlighting the crucial role of aerosol effects in shaping photochemical $O_3$ production. Under carbon neutrality–driven
emission reduction scenarios, $O_3$ exhibited pronounced seasonally contrasting responses. In winter, $O_3$ increased
monotonically with the magnitude of emission cuts, primarily due to the weakened titration by NO and the diminished aerosol-
mediated suppression via heterogeneous chemistry and radiation attenuation. In contrast, summer $O_3$ consistently declined,
with the most substantial improvements observed in high-emission urban corridors. These reductions were mainly driven by
the synergistic control of NOx and VOCs under NOx-limited and transitional photochemical regimes. When aerosol effects
were considered, wintertime $O_3$ increased by 6.7% and 10.7% under carbon peaking and neutrality scenarios, respectively,
whereas summertime $O_3$ decreased by 2.9% and 6.7%, highlighting the critical role of multiphase aerosol effects in shaping
future air quality outcomes and making climate mitigation strategies.
While this study provides innovative and policy-informative findings, several uncertainties remain that warrant further
investigation. Uncertainties primarily arise from limitations in the parameterization of heterogeneous chemistry, assumptions
in future emission projections, and the current resolution of VOCs emission inventories. Future efforts should prioritize the
enhancement of real-time VOCs monitoring, vertical profiling of $O_3$ and its precursors, and the refinement of multiphase
chemical processes in regional models. In conclusion, a holistic and mechanism-informed approach—one that jointly accounts
for emissions, aerosol effects, atmospheric chemistry, and meteorology—is essential for the effective co-control of $PM_{2.5}$ and
$O_3$ in the carbon neutrality era. Seasonally adaptive, region-specific, and chemically targeted policies are critical to maximizing
air quality and climate co-benefits under evolving environmental and policy contexts.

**Code availability**

The WRF-Chem model (version 3.7.1) used in this study is based on the standard release from NCAR (https://doi.org/10.5065/D6MK6B4K), with modifications to the aerosol and chemical mechanisms. Details of these modifications are documented in Section 2.2 of the paper. The updated code about model and NCL scripts used for data processing and visualization can be provided upon request.

**Data availability**

The FNL (Final Analysis) meteorological data are available from the Research Data Archive of NCAR: http://rda.ucar.edu/datasets/ds083.2/. The MEIC v1.4 emission inventory can be accessed at: http://meicmodel.org/?page_id=560. Hourly surface $O_3$ observations are provided by the China National Environmental Monitoring Centre (CNEMC) and are available at: http://www.cnemc.cn/.

**Author contributions**

**YL, and TW** formulated the research, and **YL**: carried it out. **ML, YQ, HW,** and **MX**: technical support on the WRF-Chem model. **CL, YL, and YW**: reviewed the manuscript.

**Competing interests**

The corresponding author has stated that all the authors have no conflicts of interest.

**Disclaimer**

Publisher's note: Copernicus Publications remains neutral about jurisdictional claims in published maps and institutional affiliations.

**Financial support**

This investigation was supported by the National Key Basic Research & Development Program of China (2024YFC3711905), the Doctoral Scientific Research Fund of Henan Finance University (2024BS055), and the National Natural Science Foundation of China (42477103), the Creative talent exchange program for foreign experts in the Belt and Road countries, the Henan Provincial Science and Technology Research and Development Program (252102320085).

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
