# Peer review of "Decadal Evolution of Aerosol-Mediated Ozone Responses in Eastern"

_EGUsphere, 2025_

## Referee Comment (RC2)

This manuscript presents a comprehensive modeling study that evaluates the driving factors controlling aerosol impacts on surface ozone  $(O_3)$  response, across two seasons (winter vs summer), to two emission reduction phases that have strategic policy shifts. By separating aerosol effects into aerosol-radiative interactions (ARI) and heterogeneous chemistry (HET), the authors show that summertime O3 responses are primarily HETdriven, while wintertime responses are mainly driven by ARI. The study also demonstrates how meteorological variability contributes to summertime O3 responses and projects how these processes may behave under air-quality control strategies. The topic is timely and of clear scientific and societal significance: it advances understanding of the nonlinear nature of photochemical O3 production and multi-pathway effects of aerosol on this process. It is also of societal significant as the conclusion is informative and understanding the driving factors will help guide emission reduction policy to be more effective and comprehensive. The modeling approach is generally appropriate and carefully implemented. However, the manuscript would benefit from major revisions to improve clarity and to remove ambiguous or potentially misleading wording. I recommend major revision; my detailed comments follow.

**Major comments:**

- 1. The manuscript alternates between two different pairwise comparisons (a) anthropogenic emissions vs. meteorological variability, and (b) aerosol-radiative interactions (ARI) vs. heterogeneous chemistry (HET) without clearly stating how these four factors relate to each other. This creates a sense of disconnection in the abstract lines 19-24, the reader sees that "anthropogenic emissions and meteorology dominate winter and summer O3, respectively" immediately followed by a discussion of ARI vs HET. Please clarify and explicitly state the conceptual framework that links the four factors.
  - In the Abstract, add a short sentence that explains the two comparisons used, like "we separate changes in O3 into those driven directly by emissions/meteorology and those mediated by aerosol processes", or after the sentence "anthropogenic emissions and meteorological variability respectively dominated winter and summer O3 increases" (line 19), follow immediately with a short clarifying sentence linking that conclusion to the ARI/HET result.
  - In the Introduction, define the four factors and their roles: anthropogenic emissions and meteorological variability are external drivers that change precursor concentrations and transport; ARI and HET are aerosol-mediated

- mechanisms that modify photochemistry and how these mechanisms mediate O3 response to precursor (NOx) decrease or meteorological variabilities.
- In the Results or discussion sections, organize the presentation so that readers
  first see the partitioning of O3 responses into contributions from emission
  reduction vs meteorology variability, and then for the emission-driven portion –
  show how aerosol processes (ARI and HET) modulate the response. Could add a
  schematic to make the logic explicit.
- 2. The Abstract's wording (Lines 22–24) that frames the Phase I–Phase II change in terms of "radical scavenging" is misleading and risks oversimplifying heterogeneous chemistry (HET). Radical uptake by aerosol (i.e., HO2 scavenging) is a loss pathway for radicals: a reduction in aerosol mass or aerosol liquid water will generally reduce this loss and therefore tends to *promote* ozone formation. Thus the statement that the "weakening of this effect during Phase II reduced O3" is unclear: if the radical-scavenging loss decreases further in Phase II, that would not by itself explain a reduction in O3. Instead, the reversal in the net HET effect between Phase I and Phase II likely reflects changes in the *net balance of multiple heterogeneous pathways* (for example, reduced radical uptake *and* changes in aerosol-mediated production or recycling of reactive nitrogen species such as HONO or ClNO2), together with changes in aerosol liquid water content and the magnitude of aerosol reductions.
  - Reword the Abstract lines 22-24 to avoid implying that radical scavenging alone explains the Phase I → Phase II sign change.
  - In the Result section 3.2, when discussing HET roles, include discussion that separates HET into its component effects: radical scavenging, heterogeneous production of reactive nitrogen like HONO and ClNO2, or at least a discussion of the chemical mechanisms used in the model parameterization of heterogeneous chemistry. In addition, a chemical diagnostics for the ozone production/loss terms during phase I and Phase II could also be useful as this allows readers to see which HET component could explain the change of sign of HET impact between phases.
- 3. The manuscript correctly notes that uncertainties in heterogeneous chemistry parameterizations could influence the results, but the current treatment is not stated and does not make clear how robust the paper's conclusions are to variations in those parameterizations. Again, HET processes directly modulate

reactive-nitrogen recycling (e.g., HONO formation,  $N_2O_5$  hydrolysis), radical budgets (HO $_2$  uptake), and hence  $O_3$  production regimes; therefore, more explicit discussion and, where possible, quantification of the uncertainty introduced by HET assumptions is essential. By adding discussion of HET impact with more details, it would help.

4. Choice of  $O_3$  metric (daily mean vs MDA8): The authors justify using daily mean  $O_3$  on the grounds that MDA8 "may underestimate full-day aerosol effects." I disagree that daily mean is a superior diagnostic for separating daytime vs nighttime processes: opposite-signed changes during day and night can cancel in the 24-h mean, obscuring mechanism interpretation. Therefore, it would help if the authors provided mean diurnal cycles of  $O_3$  (and key chemical drivers such as  $P(O_3)/L(O_3)$ ) for baseline and each phase. These plots will (i) show whether daytime and nighttime responses compensate, (ii) allow comparison with observations for model evaluation, and (iii) improve mechanistic attribution.

**Minor comments:**

- 5. Figure 3 caption panel references need correction. The panels currently cite (b) and (c) which doesn't match the description, please correct.
- 6. Figures S7-S8: the y-axis is labeled " $O_3$ ", but plotted quantity is the change in  $O_3$ , please relabel to change of  $O_3$  ( $\Delta O_3$ ).
- 7. The manuscript contains several sentences that are unclear and would benefit from careful English editing. For example, the sentence: "Therefore, the commonly used MDA8 O3 may underestimate full-day aerosol effects." is ambiguous. If the intended meaning is that using only MDA8 can miss aerosol impacts that occur outside the daytime 8-hour window, especially at night, please reword.
- 8. For figure 7, which presents changes in HO2 concentrations, it'd be clearer to express HO2 in molecules/cm^3 or ppt, as these are the standard units used for radical species. Using these units would also avoid displaying values with multiple leading zeros (as in ppb) and help readers to better assess the relative magnitude and atmospheric significance of the simulated HO2 changes.

| 9. Lines 14 – 29: the font size of Abstract does not seem consistent; lines 14-19 f size seems smaller than those of lines 20-29. | ont |
|-----------------------------------------------------------------------------------------------------------------------------------|-----|
|                                                                                                                                   |     |
|                                                                                                                                   |     |
|                                                                                                                                   |     |
|                                                                                                                                   |     |
|                                                                                                                                   |     |
|                                                                                                                                   |     |
|                                                                                                                                   |     |
|                                                                                                                                   |     |

---

## Author Comment (AC1)

**Response to Anonymous Referee #1**

**MS No.: egusphere-2025-4017**

**Title: Decadal Evolution of Aerosol-Mediated Ozone Responses in Eastern China under Clean Air Actions and Carbon Neutrality Policies**

The manuscript presents a timely modeling study on the crucial yet complex role of aerosol effects (AEs) in shaping ozone ($O_3$) trends over the Yangtze River Delta (YRD) region. The authors employ the enhanced WRF-Chem framework to explicitly and separately quantify the impacts of aerosol-radiation interactions (ARI) and heterogeneous chemistry (HET) across different policy phases and seasons, and project their influence under future carbon neutrality scenarios.

The topic is of high scientific and policy relevance, given the persistent $O_3$ pollution in China amidst successful $PM_{2.5}$ reduction. The study is well-designed, with a rigorous experimental setup (SET1-SET3) that effectively disentangles the contributions of emissions, meteorology, and aerosol processes. The findings, particularly the seasonally contrasting mechanisms (ARI-dominated in winter vs. HET-dominated in summer) and the potential for unintended $O_3$ increases from $PM_{2.5}$/NOx reductions under AEs, are novel and provide valuable insights for future air quality management. The manuscript is generally well organized and written. I recommend the manuscript for publication after minor revisions. Specific comments are listed below.

**Response:** We sincerely appreciate the reviewer's thorough evaluation and constructive comments. We have thoroughly revised the manuscript in accordance with these suggestions, which have substantially improved the quality and clarity of the work. Detailed responses to each comment are provided below, with all page and line numbers referring to the clean revised version of the manuscript.

1. While the manuscript refers to previous validation studies, it would be helpful to include at least one summary table or figure comparing observed and simulated $O_3$ (and/or key meteorological variables) for the current study period and region. This addition would improve the transparency and completeness of the paper, especially for readers who may not be familiar with the authors' earlier work.

**Response:** We sincerely thank the reviewer for this valuable suggestion. Although the performance of our WRF-Chem configuration has been validated in detail in our previous study (Li et al., 2024a), we agree that providing an explicit model–observation comparison for the specific study period will enhance the transparency and completeness of the manuscript. Accordingly, we have added a new validation table in the Supplement (Table S2), summarizing the model performance for $PM_{2.5}$, $O_3$, and key meteorological variables (2-m temperature, relative humidity, and 10-m wind speed). The table reports mean bias (MB), normalized mean bias (NMB), and correlation coefficient (R) based on observations from the national air-quality and meteorological monitoring networks across the Yangtze River Delta. The results show that the model

captures the seasonal and diurnal variability of $O_3$ and meteorological parameters with satisfactory statistical performance. A brief summary of these evaluation results has also been added to Section 3 of the revised manuscript.

**Newly added table in supplement:**

Table S2. Averaged model performance of $T_2$, $RH_2$, $WS_{10}$, $PM_{2.5}$ and $O_3$ in YRD.

| Parameter | Season | Month | MB[a] | NMB[b]/% | R[c] |
|---|---|---|---|---|---|
| $T_2$ (℃) | Summer | Jul | -0.03 | -0.08 | 0.82 |
| | Winter | Jan | 0.25 | 3.76 | 0.78 |
| $RH_2$ (%) | Summer | Jul | -1.26 | -1.89 | 0.57 |
| | Winter | Jan | -1.74 | -1.99 | 0.69 |
| $WS_{10}$ (m/s) | Summer | Jul | 0.58 | 16.88 | 0.64 |
| | Winter | Jan | 0.77 | 20.32 | 0.78 |
| $PM_{2.5}$ ($\mu g/m^3$) | Summer | Jul | -1.75 | -4.85 | 0.74 |
| | Winter | Jan | -4.34 | -8.36 | 0.63 |
| $O_3$ (ppb) | Summer | Jul | 1.54 | 5.33 | 0.66 |
| | Winter | Jan | -5.26 | -14.02 | 0.58 |

MB[a]: mean bias; NMB[b]: normal mean bias; R[c]: correlation coefficient.

**Manuscript changes (Section 3, Page 10, lines 222-226):**

"The accuracy of simulated meteorological parameters and pollutant concentrations under scenario (20E20M_AEs) has been thoroughly validated against ground-based observations in earlier work (Li et al., 2024a). As summarized in Table S2, the model reasonably captures the magnitude, seasonal variability of $PM_{2.5}$, $O_3$, as well as the major features of temperature, relative humidity, and wind speed. These results provide confidence in the model's ability to represent the atmospheric conditions relevant to the subsequent analysis."

**References:**

Li, Y., Wang, T., Wang, Q. g., Li, M., Qu, Y., Wu, H., and Xie, M.: Exploring the role of aerosol-ozone interactions on O3 surge and PM2.5 decline during the clean air action period in Eastern China 2014–2020, Atmos. Res., 302, 107294, https://doi.org/10.1016/j.atmosres.2024.107294, 2024a.

2. The assumption of proportional reductions (10–90%) across all pollutants is understandable for simplicity but may not fully capture realistic sectoral differences in future emission pathways. Please discuss this limitation and, if possible, comment on how it might influence the overall conclusions.

**Response:** We thank the reviewer for this valuable comment. We agree that applying proportional reductions to all pollutants is a simplification that does not fully represent sector-specific emission trajectories under carbon neutrality policies. This assumption was adopted mainly to maintain a consistent and comparable framework for evaluating the nonlinear $O_3$ responses to precursor reductions and aerosol effects, rather than to reproduce specific policy pathways. We acknowledge that such heterogeneity may

influence the magnitude of O$_3$ responses and modify the relative contributions of different precursor groups. This limitation has now been explicitly discussed in the revised manuscript. In future work, we plan to incorporate sector-resolved, scenario-specific emission pathways to better represent realistic emission evolution and to further assess how these structural differences may modulate O$_3$ sensitivity.

**Manuscript changes (Section 3.4, Page 23, lines 485-491):**

"The proportional 10-90% reductions applied uniformly across all pollutant species were designed as an idealized framework to systematically examine nonlinear O$_3$ responses under consistent boundary conditions. In practice, however, future emission pathways are expected to exhibit pronounced sectoral and spatial heterogeneity—for example, SO$_2$ and primary PM$_{2.5}$ typically decline faster than VOCs and NH$_3$, and the pace of reductions varies across industrial, transportation, and residential sectors. Such differences may influence the magnitude of O$_3$ responses and the balance among precursor contributions. Recognizing this limitation, future work will incorporate sector-resolved and scenario-specific emission pathways to provide a more realistic assessment of O$_3$ sensitivity under evolving emission structures."

3. From Sections 3.1 to 3.4, please clarify how the mean pollutant concentrations were calculated—are they spatial grid averages or site-based averages? This information is important for interpreting the representativeness of spatial and temporal trends.

**Response:** Thanks for the question. The calculation of average pollutants concentration in Sections 3.1 to 3.4 was based on the grid average within the specified region.

4. It seems that the effects of ARI and HET are independent, i.e., there may be nonlinear interaction between the two effects. This should be noted and discussed.

**Response:** We thank the reviewer for raising this insightful point. We agree that aerosol–radiation interactions (ARI) and heterogeneous chemistry (HET) are not strictly independent and that nonlinear interactions between them may occur. To address this, we have added a dedicated paragraph in the revised manuscript.

**Manuscript changes (Section 3.2, Pages 17-18, lines 357-367):**

"Previous studies showed that ARI and HET were not fully independent and could interact through aerosol–meteorology–chemistry feedbacks (Chen et al., 2019; Liu et al., 2023b; Kong et al., 2018; Li et al., 2020a). ARI-induced increases in near-surface relative humidity typically enhanced aerosol hygroscopic growth and expanded aerosol surface area. The resulting increase in aerosol liquid water promoted gas-to-particle partitioning and facilitated aqueous- and surface-phase reactions, thereby accelerating heterogeneous oxidation pathways involving SO$_2$ and NOx. The strengthened heterogeneous formation of secondary inorganic aerosols further modified solar radiation and potentially intensified the ARI effect. In the present study, our primary focus was to quantify the separate and combined contributions of ARI and HET to O$_3$ changes across different stages of the CAAP. Accordingly, we isolated their individual

impacts rather than examining their nonlinear coupling. We acknowledged that ARI–HET interactions might also affect $O_3$ under certain chemical and meteorological conditions, and we indicated that future work would incorporate dedicated coupled-sensitivity experiments to more explicitly quantify these nonlinearities and their implications for $O_3$ formation."

**References:**

Chen, J., Li, Z., Lv, M., Wang, Y., Wang, W., Zhang, Y., Wang, H., Yan, X., Sun, Y., and Cribb, M.: Aerosol hygroscopic growth, contributing factors, and impact on haze events in a severely polluted region in northern China, Atmos. Chem. Phys., 19, 1327-1342, https://doi.org/10.5194/acp-19-1327-2019, 2019.

Kong, L., Du, C., Zhanzakova, A., Cheng, T., Yang, X., Wang, L., Fu, H., Chen, J., and Zhang, S.: Trends in heterogeneous aqueous reaction in continuous haze episodes in suburban Shanghai: an in-depth case study, Sci. Total Environ., 634, 1192-1204, https://doi.org/10.1016/j.scitotenv.2018.04.086, 2018.

Li, J., Han, Z., Li, J., Liu, R., Wu, Y., Liang, L., and Zhang, R.: The formation and evolution of secondary organic aerosol during haze events in Beijing in wintertime, Sci. Total Environ., 703, 134937, https://doi.org/10.1016/j.scitotenv.2019.134937, 2020a.

Liu, Z., Wang, H., Peng, Y., Zhang, W., Che, H., Zhang, Y., Liu, H., Wang, Y., Zhao, M., Zhang, X. The combined effects of heterogeneous chemistry and aerosol-radiation interaction on severe haze simulation by atmospheric chemistry model in Middle-Eastern China. Atmos. Environ. 302, 119729. https://doi.org/10.1016/j.atmosenv.2023.119729, 2023b.

5. Line175: Table 1 contains typographical issues: several entries for "10% reduction" appear where "40%, 60%, 80%" were intended—please correct.

**Response:** We sincerely thank the reviewer for noticing this error. After careful checking, we confirm that the ratios listed as 10% for CUT_MEIC_40, CUT_MEIC_60, and CUT_MEIC_80 in Table 1 were a writing error. The correct reduction ratios have now been revised in Table 1 in the updated manuscript (**Page 8, line 181**).

6. Line 445: Please clarify what specific "carbon neutrality–aligned emission trajectories" are referred to here. Is it the specific 50% reduction scenario, or a broader set of pathways?

**Response:** We thank the reviewer for pointing out this ambiguity. In the revised manuscript, we have clarified that the term "carbon-neutrality–aligned emission trajectories" refers specifically to the proportional multi-pollutant reduction pathways (10–90%) used in this study. These pathways are not intended to represent a single policy scenario such as the 50% reduction case; rather, they serve as stylized, economy-wide emission decline trajectories consistent with the long-term direction required for carbon neutrality. The revised text now explicitly states this definition to avoid

misunderstanding.

**Manuscript changes (Section 3.4, Page 22, lines 470-473):**

"These results underscore the seasonal asymmetry of $O_3$ responses under the carbon-neutrality–aligned emission trajectories used in this study—namely the proportional precursor-reduction pathways designed to reflect long-term, economy-wide emission declines. While such stringent reductions may inadvertently aggravate wintertime $O_3$ pollution, they yield substantial co-benefits for summer $O_3$ mitigation."

7. Ensure consistent use of "Clean Air Action Plan (CAAP)" throughout the manuscript. Avoid alternating between "CAAP" and "Clean Air Action Plan" in figure captions and text for terminological uniformity.

**Response:** We thank the reviewer for this helpful comment. We have carefully checked the entire manuscript, including all figure captions and supplementary materials, and ensured consistent use of the term "Clean Air Action Plan (CAAP)" throughout. Instances where "Clean Air Action Plan" or mixed forms previously appeared have now been corrected for terminological uniformity.

8. Please consistently use "VOCs" (plural) when referring to volatile organic compounds.

**Response:** We thank the reviewer for pointing this out. We have thoroughly checked the entire manuscript, including the main text, figures, and supplementary materials, and have now corrected all instances of "VOC" to the consistent plural form "VOCs" when referring to volatile organic compounds.

9. English of the manuscript needs to be improved.

**Response:** We thank the reviewer for this helpful suggestion. The manuscript has been thoroughly revised to improve clarity, grammar, and overall readability, with particular attention paid to the **Methods and Results** sections, where technical descriptions and interpretations have been carefully polished and refined. In addition to our own revisions, the manuscript has undergone an additional round of professional-level language editing. We believe these revisions have substantially improved the fluency and clarity of the manuscript.

*We would like to once again express our sincere gratitude to the reviewers for their thoughtful and constructive comments. Their insights have been invaluable and have greatly enhanced the clarity, rigor, and overall quality of our manuscript.*

---

## Author Comment (AC2)

**Response to Anonymous Referee #2**

**MS No.: egusphere-2025-4017**

**Title: Decadal Evolution of Aerosol-Mediated Ozone Responses in Eastern China under Clean Air Actions and Carbon Neutrality Policies**

This manuscript presents a comprehensive modeling study that evaluates the driving factors controlling aerosol impacts on surface ozone ($O_3$) response, across two seasons (winter vs summer), to two emission reduction phases that have strategic policy shifts. By separating aerosol effects into aerosol-radiative interactions (ARI) and heterogeneous chemistry (HET), the authors show that summertime $O_3$ responses are primarily HET- driven, while wintertime responses are mainly driven by ARI. The study also demonstrates how meteorological variability contributes to summertime $O_3$ responses and projects how these processes may behave under air-quality control strategies. The topic is timely and of clear scientific and societal significance: it advances understanding of the nonlinear nature of photochemical $O_3$ production and multi-pathway effects of aerosol on this process. It is also of societal significant as the conclusion is informative and understanding the driving factors will help guide emission reduction policy to be more effective and comprehensive. The modeling approach is generally appropriate and carefully implemented. However, the manuscript would benefit from major revisions to improve clarity and to remove ambiguous or potentially misleading wording. I recommend major revision; my detailed comments follow.

**Response:** We sincerely appreciate the reviewer's thorough evaluation and constructive comments. We have thoroughly revised the manuscript in accordance with these suggestions, which have substantially improved the quality and clarity of the work. Detailed responses to each comment are provided below, with all page and line numbers referring to the clean revised version of the manuscript.

**Major comments:**

1. The manuscript alternates between two different pairwise comparisons - (a) anthropogenic emissions vs. meteorological variability, and (b) aerosol-radiative interactions (ARI) vs. heterogeneous chemistry (HET) - without clearly stating how these four factors relate to each other. This creates a sense of disconnection in the abstract lines 19-24, the reader sees that "anthropogenic emissions and meteorology dominate winter and summer $O_3$, respectively" immediately followed by a discussion of ARI vs HET. Please clarify and explicitly state the conceptual framework that links the four factors.

**Response:** We sincerely thank the reviewer for this insightful and constructive comment. We agree that the manuscript required a clearer articulation of the conceptual framework linking the four factors—anthropogenic emissions, meteorological variability, ARI, and HET. Following the reviewer's suggestions, we have revised the

Abstract, Introduction, and the structure of the Results section to explicitly clarify their hierarchical relationships. Below we summarize our revisions for each sub-point.

(1) In the Abstract, add a short sentence that explains the two comparisons used, like "we separate changes in $O_3$ into those driven directly by emissions/meteorology and those mediated by aerosol processes", or after the sentence "anthropogenic emissions and meteorological variability respectively dominated winter and summer $O_3$ increases" (line 19), follow immediately with a short clarifying sentence linking that conclusion to the ARI/HET result.

**Response:** We thank the reviewer for this constructive suggestion. As suggested, we added a concise clarifying sentence to explicitly link the two dimensions of comparison.

**Abstract changes (Page 2, lines 19-20):**

"We separate $O_3$ changes into those driven directly by anthropogenic emissions and meteorological variability, and those mediated by aerosol processes through ARI and HET."

(2) In the Introduction, define the four factors and their roles: anthropogenic emissions and meteorological variability are external drivers that change precursor concentrations and transport; ARI and HET are aerosol-mediated mechanisms that modify photochemistry and how these mechanisms mediate $O_3$ response to precursor (NOx) decrease or meteorological variabilities.

**Response:** We thank the reviewer for this constructive suggestion. In the Introduction, we added a new paragraph that defines the four factors and clarifies their hierarchical roles.

**Manuscript changes (Section 1, Page 4, lines 72-77):**

"Anthropogenic emissions and meteorological variability act as external drivers that directly regulate precursor concentrations, atmospheric chemical regimes, and transport processes. In contrast, ARI and HET represent aerosol-mediated mechanisms that reshape the photochemical environment by altering photolysis rates and radical budgets. These aerosol-driven mechanisms determine the extent to which surface $O_3$ responds to precursor (particularly NOx) reductions or meteorological perturbations. This conceptual framework underpins our separation of $O_3$ changes into externally driven components and aerosol-modulated components in this study."

(3) In the Results or discussion sections, organize the presentation so that readers first see the partitioning of $O_3$ responses into contributions from emission reduction vs meteorology variability, and then – for the emission-driven portion – show how aerosol processes (ARI and HET) modulate the response. Could add a schematic to make the logic explicit.

**Response:** We thank the reviewer for this constructive suggestion. The manuscript already follows this structure: Section 3.1 quantifies the contributions of anthropogenic emission reductions and meteorological variability to $O_3$ changes, and Sections 3.2

evaluate how ARI and HET further modulate these externally driven O₃ responses. To make this hierarchical relationship clearer, we have added explicit transition sentences at the end of Section 3.1 and at the beginning of Sections 3.2, emphasizing that ARI and HET act as aerosol-mediated modifiers of the emission-driven O₃ changes.

**Manuscript changes:**

**(Section 3.1, Page 11, lines 259-260)**: "These externally driven O₃ changes provide the foundation for evaluating how aerosol-mediated processes further modulate the emission-reduction-driven portion of the O₃ response."

**(Section 3.2, Page 12, lines 266-267)**: "Building on the external drivers identified in Section 3.1, we next examined how ARI and HET modified the emission-reduction-driven O₃ response."

In addition, we have included a simple schematic in the supplement and a description in Results section illustrating the conceptual framework linking external drivers and aerosol-mediated processes, which helps clarify the logic of the analysis. These revisions collectively clarify the conceptual structure and improve readability.

**Manuscript changes (Section 3.2, Page 18, lines 368-371):**

"Figure S8 illustrated the hierarchical relationships among the four factors analyzed in this section. Emission reductions and meteorological variability constituted the external drivers of O₃ changes, whereas ARI and HET acted as aerosol-mediated modulators that adjust the emission-reduction-driven O₃ responses. This framework motivated our presentation sequence, where external drivers were examined first, followed by the modulation effects of ARI and HET."

**Newly added figure in supplement:**

[Figure]

Figure S8. Schematic overview of the analytical framework separating externally driven O₃ changes (from emissions and meteorology) from the aerosol-mediated modulation by ARI and HET, which jointly determine the seasonal and phase-dependent O₃ responses.

2.  The Abstract's wording (Lines 22–24) that frames the Phase I–Phase II change in terms of "radical scavenging" is misleading and risks oversimplifying heterogeneous chemistry (HET). Radical uptake by aerosol (i.e., $HO_2$ scavenging) is a loss pathway for radicals: a reduction in aerosol mass or aerosol liquid water will generally reduce this loss and therefore tends to promote ozone formation. Thus, the statement that the "weakening of this effect during Phase II reduced $O_3$" is unclear: if the radical-scavenging loss decreases further in Phase II, that would not by itself explain a reduction in $O_3$. Instead, the reversal in the net HET effect between Phase I and Phase II likely reflects changes in the net balance of multiple heterogeneous pathways (for example, reduced radical uptake and changes in aerosol-mediated production or recycling of reactive nitrogen species such as HONO or $ClNO_2$), together with changes in aerosol liquid water content and the magnitude of aerosol reductions.

**Response:** We thank the reviewer for this important comment. We agree that describing the Phase I → Phase II change of the HET effect solely in terms of radical scavenging oversimplifies the heterogeneous chemistry represented in WRF-Chem. To address this, we have revised both the Abstract and Section 3.2.

(1) Reword the Abstract lines 22-24 to avoid implying that radical scavenging alone explains the Phase I → Phase II sign change.

**Response:** We thank the reviewer for this important comment. We agree that the previous Abstract wording unintentionally overemphasized radical scavenging and did not adequately reflect the multicomponent nature of heterogeneous chemistry. To address this concern, we revised the Abstract to clarify that the Phase I → Phase II sign reversal of the HET effect arises from the combined influence of several heterogeneous pathways—not radical uptake alone.

**Abstract changes (Page 2, lines 23-26):**

   "Summer $O_3$ was more sensitive to HET. In Phase I, aerosol decreases weakened heterogeneous radical uptake, enhancing $O_3$ formation (+1.62 ppb). In Phase II, however, the net HET effect reversed sign (–2.86 ppb), driven by shifts in multiple heterogeneous pathways—including changes in radical uptake, HONO and $N_2O_5$ chemistry, and aerosol liquid water—rather than radical scavenging alone."

(2) In the Result section 3.2, when discussing HET roles, include discussion that separates HET into its component effects: radical scavenging, heterogeneous production of reactive nitrogen like HONO and $ClNO_2$, or at least a discussion of the chemical mechanisms used in the model parameterization of heterogeneous chemistry. In addition, a chemical diagnostics for the ozone production/loss terms during phase I and Phase II could also be useful as this allows readers to see which HET component could explain the change of sign of HET impact between phases.

**Response:** We greatly appreciate the reviewer's valuable suggestions. In response, both Section 2.2 and Section 3.2 have been substantially revised to provide a clearer and more mechanism-based presentation of heterogeneous chemistry (HET) in WRF-Chem.

Section 2.2 has been expanded to introduce, for the first time in this study, the full set of heterogeneous reaction pathways and the updated HET module implemented in our model (detailed descriptions of all HET pathways and parameterizations are provided in Table S1 (response to comment 3)). The revised manuscript now explicitly describes the major heterogeneous pathways represented in the model, including: (1) radical uptake ($HO_2$, OH, $NO_3$), (2) $NO_2$ heterogeneous conversion to HONO and $HNO_3$, (3) $N_2O_5$ hydrolysis regulating nighttime NOx partitioning, (4) $SO_2$ and $H_2SO_4$ heterogeneous oxidation, and (5) direct $O_3$ uptake on dust and black carbon surfaces.

Section 3.2 then builds on this framework to explain how the relative contributions of these pathways differ between Phase I and Phase II and how they lead to opposite $O_3$ responses. Although a full integrated process rate (IPR) analysis was not available for this study, the added diagnostics of $HO_2$, HONO, and $N_2O_5$ provide direct and independent evidence supporting the heterogeneous pathways responsible for the sign reversal. The revised discussion therefore offers a mechanistic, process-consistent, and evidence-supported explanation for the observed transition from a positive HET effect in Phase I to a negative effect in Phase II. In subsequent studies, we plan to incorporate a full IPR framework, employ more explicit radical and reactive-nitrogen diagnostics, and further refine the representation of aerosol liquid water and heterogeneous reaction parameterizations—all of which will help strengthen the process attribution and deepen the mechanistic understanding of aerosol-mediated $O_3$ responses.

**Manuscript changes:**
**(Section 2.2, Page 6, lines 127-134):**
"Heterogeneous chemistry exerts complex influences on $O_3$ formation by altering radical budgets, modifying reactive nitrogen cycling, and changing aerosol-phase reaction rates. In the enhanced WRF-Chem, HET is represented through multiple pathways on dust and black carbon surfaces, including (1) heterogeneous uptake of $HO_2$, OH, $NO_2$, and $NO_3$; (2) nighttime $N_2O_5$ hydrolysis to $2HNO_3$; (3) heterogeneous formation of HONO from $NO_2$ uptake on carbonaceous aerosols; (4) $SO_2$ and $H_2SO_4$ heterogeneous oxidation; and (5) direct $O_3$ uptake on dust and black carbon surfaces. These processes collectively modify photolysis-driven radical initiation and NOx partitioning. Therefore, the net HET effect reflects the balance among several aerosol-mediated pathways rather than a single mechanism. The specific heterogeneous reactions and their corresponding uptake coefficients ($\gamma$) used in this study are listed in Table S1."
**(Section 3.2, Pages 15-16, lines 313-339):**
"During Phase I, the substantial reductions in aerosol mass and surface area primarily weakened $HO_2$ heterogeneous uptake, as indicated by elevated $HO_2$ (Figure 7d). This reduction in radical loss increased the availability of $HO_2$ and OH, leading to an enhancement in the photochemical ozone production term $P(O_3)$ (Dyson et al., 2023). In parallel, $N_2O_5$ also increased during Phase I (Figure S5a), consistent with suppressed heterogeneous hydrolysis under reduced aerosol liquid water (ALW) and diminished aerosol surface area (Brown and Stutz, 2012). The weakened $N_2O_5$ hydrolysis further

limited nighttime conversion of reactive nitrogen to $HNO_3$, maintaining NOx in more photochemically active forms (Ma et al., 2023b). Meanwhile, heterogeneous $NO_2$ uptake—an important HONO source—was significantly reduced, consistent with the simulated decrease in HONO (Figure S5d). The reduction in HONO slightly weakened early-morning radical initiation (Yu et al., 2022), but this influence was outweighed by the strong enhancement in $HO_2$ and the limited conversion of NOx into $HNO_3$. As a result, HET exerted a net positive contribution to $O_3$ (+1.62 ppb) in Phase I.

In contrast, Phase II exhibited a fundamentally different chemical response. Although aerosol loadings continued to decrease, the relative importance of heterogeneous pathways shifted substantially. $HO_2$ declined during Phase II (Figure 7d), indicating a reduced radical pool and weaker propagation of daytime photochemical production. At the same time, $N_2O_5$ decreased markedly (Figure S5b), suggesting that nighttime $NO_3/N_2O_5$ chemistry became less effective at sustaining reactive nitrogen cycling under even lower aerosol surface area and ALW. Rather than promoting efficient nighttime NOx recycling, this suppression favored a net loss of reactive nitrogen through terminal sinks (e.g., $HNO_3$), shifting NOx partitioning toward less photochemically active forms and weakening daytime $P(O_3)$. Conversely, HONO concentrations rebounded during Phase II (Figure S5e). This increase reflects the altered balance between $NO_2$ uptake and nighttime NOx partitioning under reduced $N_2O_5$ hydrolysis. However, despite this HONO increase, its positive effect on radical initiation could not compensate for the combined decline in $HO_2$, weakened $N_2O_5$ hydrolysis, and enhanced $HNO_3$ formation (George et al., 2015). The joint effect was a net reduction in the morning radical pool and diminished photochemical $O_3$ production (-2.86 ppb). This multi-pathway adjustment explains the observed sign reversal of HET's effect on $O_3$ between the two phases and underscores the importance of considering the full suite of heterogeneous processes—rather than radical uptake alone—when interpreting aerosol-mediated $O_3$ responses. In future work, we plan to apply integrated process rate (IPR) diagnostics to more directly evaluate how individual heterogeneous pathways—such as $HO_2$ uptake, HONO formation, and $N_2O_5$ hydrolysis—shape the resulting $O_3$ responses. Coupled with continued improvements in heterogeneous chemistry parameterizations and more comprehensive constraints on radical, reactive nitrogen, and aerosol liquid water fields, this will enable a more detailed and process-resolved understanding of phase-dependent $O_3$ changes."

**Newly added figure in supplement:**

[Figure]

Figure S5 Spatial distributions of N$_2$O$_5$ (a-c) and HONO (d-f) concentration (ppb) changes induced by aerosol heterogeneous chemistry (HET) in summer during two phases of the Clean Air Action in the Yangtze River Delta (YRD).


**Response:** We sincerely thank the reviewer for raising this important point. We agree that uncertainties in the parameterization of heterogeneous chemistry (HET)—including the uptake coefficients and their dependencies on aerosol liquid water content, acidity, and particle composition—may influence the simulated $O_3$ responses. We acknowledge that this constitutes an important limitation of the current study and appreciate the opportunity to clarify it more explicitly.

First, although we recognize the scientific value of quantifying these uncertainties through targeted sensitivity simulations, the present work was designed as a multi–phase, multi–season modeling framework that prioritizes consistency across scenarios. Systematically perturbing heterogeneous uptake coefficients would require rerunning the entire experimental suite under multiple alternative chemical configurations, which falls outside the methodological scope and computational design of this study. We have now explicitly acknowledged this limitation in the revised manuscript and clarified why such experiments, while valuable, were not included in the current modeling framework.

Second, in line with the reviewer's helpful suggestion, we have expanded the Supplementary Material (Table S1) to provide a more detailed and transparent description of the heterogeneous pathways newly implemented in our WRF-Chem configuration, particularly those occurring on dust and black carbon surfaces. These additions include the corresponding parameterizations and supporting references, thereby improving clarity and reproducibility.

Third, Section 3.5 (Discussion) has been substantially revised to include a more explicit examination of uncertainties associated with $HO_2$, $NO_2$, and $N_2O_5$ uptake coefficients. We further discuss how variations in these parameters could modify radical budgets, reactive nitrogen cycling, and consequently the magnitude of $O_3$ responses. While the absolute perturbations may vary under different plausible $\gamma$ values, the phase-dependent sign reversal of the HET effect is unlikely to change, as it arises from a structural shift in the relative importance of multiple heterogeneous pathways rather than the sensitivity of any single reaction. This robustness is also supported by previous studies (e.g., Shao et al., 2021; Li et al., 2019), which show that different $HO_2$ uptake coefficients alter the magnitude but not the direction of the $O_3$ response.

Finally, we fully agree with the reviewer on the importance of more rigorous

characterization of HET-related uncertainties. In future work, we plan to incorporate dedicated heterogeneous-chemistry sensitivity simulations together with integrated process rate (IPR) diagnostics to quantitatively evaluate the contribution and robustness of individual HET pathways. These improvements will help further reduce uncertainty and strengthen the mechanistic interpretation of aerosol–$O_3$ interactions under evolving emission scenarios.

We sincerely appreciate the reviewer's constructive comments, which have allowed us to provide clearer documentation of the heterogeneous chemistry processes included in this study and a more comprehensive evaluation of their associated uncertainties.

**Newly added table in supplement:**

Table S1. The heterogeneous reactions and the uptake coefficients are considered in our study.

| Reaction | Uptake Coefficient $\gamma$ | Reference |
|---|---|---|
| **(a) dust** | | |
| $O_3$ (g) → $O_3$ (ads) | $1\times10^{-4}$ | (Bauer et al., 2004) |
| OH (g) → OH (ads) | $\dfrac{0.18}{1+(RH\times100)^{0.36}}$ | (Bedjanian et al., 2013) |
| $HO_2$ (g) → $HO_2$ (ads) | 0.1 | (Phadnis and Carmichael, 2000) |
| $H_2O_2$ (g) → $H_2O_2$ (ads) | $3.33\times10^{-4}$  (RH<0.15) | (Pradhan et al., 2010) |
| | $3.33\times10^{-4}+\dfrac{(RH-0.15)\times2.7\times10^{-4}}{0.55}$ | |
| | (0.15<RH<0.7) | |
| | $6.03\times10^{-4}$  (RH>0.7) | |
| $NO_2$ (g) → $HNO_3$ (ads) | $5.0\times10^{-5}$ | (Li et al., 2019c) |
| $NO_3$ (g) → $HNO_3$ (ads) | $3.0\times10^{-3}$ | (Li et al., 2019c) |
| $HNO3$ (g) → $HNO_3$ (ads) | $1.0\times10^{-2}$ | (Liu et al., 2008) |
| $N_2O_5$ (g) →2$HNO_3$ (ads) | $3.0\times10^{-3}$  (RH<0.3) | (Bauer et al., 2004) |
| | $0.0425\times RH-0.00975$ | |
| | (0.15<RH<0.7) | |
| | $2.0\times10^{-2}$  (RH>0.7) | |
| $SO_2$ (g) →$SO_4^{2-}$ (ads) | $1.0\times10^{-4}$  (RH < 0.5) | (Zheng et al., 2015) |
| | $1.0\times10^{-4}+\dfrac{RH-0.5}{(1-0.5)\times2\times10^{-4}}$ | |
| | (RH>0.5) | |
| $H_2SO_4$(g) → $SO_4^{2-}$ (ads) | $5.0\times10^{-2}$  (RH<0.5) | (Huang et al., 2014) |
| | 0.1 (RH>0.5) | |
| **(b) Black carbon** | | |

| | | |
|---|---|---|
| $O_3$ (g) $\rightarrow$ $O_3$ (ads) | $1.8\times10^{-4}\times e^{-\frac{1000}{T}}$ | (Tie et al., 2005) |
| OH (g) $\rightarrow$ OH (ads) | $5.0\times10^{-2}$ | (Slade and Knopf, 2013) |
| $HO_2$ (g) $\rightarrow$ $HO_2$ (ads) | $1.0\times10^{-2}$ | (Saathoff et al., 2001) |
| $NO_2$(g) $\rightarrow$ 0.5HONO + 0.5HNO$_3$ | $5.0\times10^{-4}$ | (Lei et al., 2004) |
| $NO_3$ (g) $\rightarrow$ $HNO_3$ (ads) | $3.0\times10^{-4}$ (RH<0.5) | (Saathoff et al., 2001) |
| | $1.0\times10^{-4}$ (RH>0.5) | |
| $N_2O_5$ (g) $\rightarrow$ $2HNO_3$ (ads) | $4.0\times10^{-5}$ (RH<0.5) | (Saathoff et al., 2001) |
| | $2.0\times10^{-4}$ (RH>0.5) | |
| $HNO_3$ (g) $\rightarrow$ $HNO_3$ (ads) | $1.0\times10^{-3}$ | (Rogaski et al., 1997) |

mechanism interpretation. Therefore, it would help if the authors provided mean diurnal cycles of $O_3$ (and key chemical drivers such as $P(O_3)/L(O_3)$) for baseline and each phase. These plots will (i) show whether daytime and nighttime responses compensate, (ii) allow comparison with observations for model evaluation, and (iii) improve mechanistic attribution.

**Response:** We sincerely thank the reviewer for this valuable and constructive comment. We agree that relying solely on daily mean $O_3$ may mask potential compensating effects between daytime photochemical production and nighttime deposition or titration. In light of this concern, we have removed the original statement claiming that MDA8 may underestimate full-day aerosol effects, as it could be misleading without supporting diurnal diagnostics.

To directly address the reviewer's concern, we have added diurnal-cycle analyses for baseline $O_3$ and the aerosol-mediated $O_3$ changes (HET, ARI, and AEs) for Phase I, Phase II, and the overall Clean Air Action period in both winter and summer (Figures S6–S7). These results show that $O_3$ perturbations across all phases and both seasons are overwhelmingly dominated by daytime changes near the photochemical peak (14–16 LT), whereas nighttime variations are much smaller in magnitude and share the same sign as the daytime responses. Therefore, nighttime effects do not offset or compensate daytime changes. This indicates that the daily-mean $O_3$ responses presented in the main text reflect genuine daytime-dominant adjustments rather than artifacts of day–night cancellation. A new paragraph summarizing these findings has been added to Section 3.2.

Regarding the reviewer's suggestion to include diurnal variations in $P(O_3)/L(O_3)$, we greatly appreciate this recommendation. In the current modeling framework, however, the integrated process rate (IPR) module was not activated, as the experimental design focused on maintaining a consistent chemical configuration across a large suite of multi-phase and multi-season simulations. Enabling IPR would require rerunning the full experimental set under an alternative chemistry configuration, which falls outside the methodological scope of the present study. We have clarified this constraint in the revised manuscript. As an alternative, we provide diagnostics of key radical and reactive nitrogen species ($HO_2$, HONO, and $N_2O_5$), which capture the fundamental processes controlling photochemical production and nighttime nitrogen cycling, and thus offer mechanistic insight comparable to $P(O_3)/L(O_3)$. Further details on these diagnostics and their interpretation are provided in our response to Comment 2.

We sincerely thank the reviewer again for this insightful comment. The additions and clarifications prompted by this suggestion have substantially improved our manuscript by providing stronger mechanistic support and clearer justification of the chosen $O_3$ metric.

**Newly added figure in supplement:**

[Figure]

Figure S6. Diurnal variations of (a) baseline $O_3$ concentrations (ID: 20E20M_AEs) and (b–d) aerosol-induced $O_3$ changes ($\Delta O_3$) in winter. Panels show the impacts during the overall Clean Air Action period (CAAP), Phase I, and Phase II, each decomposed into heterogeneous chemistry (HET), aerosol–radiation interactions (ARI), and their combined effects (AEs).

[Figure]

Figure S7. Diurnal variations of (a) baseline $O_3$ concentrations (ID: 20E20MI_AEs) and (b–d) aerosol-induced $O_3$ changes ($\Delta O_3$) in summer. Panels show the impacts during the overall Clean Air Action period (CAAP), Phase I, and Phase II, each decomposed into heterogeneous chemistry (HET), aerosol–radiation interactions (ARI), and their combined effects (AEs).

**Manuscript changes (Section 3.2, Page 17, lines 343-356):**

"To further evaluate whether daytime and nighttime $O_3$ responses compensate within the daily mean metric, we examined the diurnal cycles of baseline $O_3$ concentration and the aerosol-mediated impacts (HET, ARI, and AEs) during Phase I, Phase II, and the overall CAAP period for both winter (Figure S6) and summer (Figure S7). Across all phases and both seasons, the dominant $O_3$ perturbations occur during daytime hours, coinciding with the photochemical peak at 14–16 LT. In winter, Phase I exhibits a pronounced daytime enhancement driven by ARI (up to ~2.41 ppb), whereas HET induces a consistently positive but comparatively weaker increase (up to ~0.49 ppb). In Phase II, the ARI-induced enhancement weakens notably (peaking at ~1.24 ppb), and HET-induced changes remain minor. In summer, the diurnal behavior more clearly reflects a daytime‐dominated response. During Phase I, HET produces a marked midday $O_3$ enhancement (up to ~2.01 ppb), while ARI imposes a weaker yet persistent negative contribution. In contrast, Phase II is characterized by a strong HET-driven daytime $O_3$ decrease (maximum ~3.43 ppb), overwhelming the comparatively modest positive ARI effect. For all cases, nighttime $O_3$ changes share the same direction as daytime responses but remain substantially smaller in magnitude, insufficient to offset the daytime signals dominated by photochemistry. These diurnal patterns confirm that the phase-dependent $O_3$ responses to aerosol effects are not artifacts of day–night compensation in daily mean metrics, but instead arise from robust, daytime-dominant photochemical adjustments."

**Minor comments:**

5. Figure 3 caption – panel references need correction. The panels currently cite (b) and (c) which doesn't match the description, please correct.

**Response:** We thank the reviewer for pointing out this oversight. The panel references in the caption of Figure 3 have now been carefully checked and corrected to ensure full consistency with the figure layout and the corresponding descriptions in the text. The revised caption accurately refers to the correct subpanels and their contents **(Page 12, lines 262-264)**.

6. Figures S7-S8: the y-axis is labeled "$O_3$", but plotted quantity is the change in O3, please relabel to change of $O_3$ ($\Delta O_3$).

**Response:** We thank the reviewer for catching this labeling error. In the revised Supplementary Information, the original Figures S7–S8 have been renumbered as Figures S11–S12. In these figures, the plotted variable represents the change in $O_3$ rather than the absolute concentration. We have therefore corrected the y-axis label to "$\Delta O_3$ (ppb)" in both figures to accurately reflect the displayed quantity. This correction is purely a labeling issue and does not affect the interpretation or conclusions of the results.

7. The manuscript contains several sentences that are unclear and would benefit from careful English editing. For example, the sentence: "Therefore, the commonly used

MDA8 O$_3$ may underestimate full-day aerosol effects." is ambiguous. If the intended meaning is that using only MDA8 can miss aerosol impacts that occur outside the daytime 8-hour window, especially at night, please reword.

**Response:** We thank the reviewer for this valuable comment. We agree that the original sentence ("Therefore, the commonly used MDA8 O$_3$ may underestimate full-day aerosol effects.") was ambiguous and could cause confusion. Our intended meaning was that reliance on MDA8 O$_3$, which represents only the maximum 8-hour daytime average, may fail to capture aerosol-related influences occurring outside this window, including early morning and nighttime periods. To avoid misinterpretation, we have removed this sentence from the revised manuscript. In addition, we have carefully reviewed and edited the surrounding text, as well as other passages identified as unclear, with particular attention to the **Methods and Results** sections, where technical descriptions and interpretations have been further refined and clarified. These revisions have improved the overall clarity, precision, and readability of the manuscript.

8. For figure 7, which presents changes in HO$_2$ concentrations, it'd be clearer to express HO$_2$ in molecules/cm^3 or ppt, as these are the standard units used for radical species. Using these units would also avoid displaying values with multiple leading zeros (as in ppb) and help readers to better assess the relative magnitude and atmospheric significance of the simulated HO$_2$ changes.

**Response:** We thank the reviewer for this helpful suggestion. We have updated Figure 7 by converting HO$_2$ concentrations from ppb to ppt. The revised figure now presents HO$_2$ in ppt, which improves readability and facilitates comparison with previous modeling and observational studies. The corresponding figure caption and text description in Section 3.2 have also been updated accordingly.

**Revised Figure 7 (Page 17, lines 340-342):**

[Figure]

Figure 7 Spatial distributions of $HO_2$ concentration (ppt) changes induced by aerosol heterogeneous chemistry (HET) in winter (a-c) and summer (d-f) during two phases of the Clean Air Action in the Yangtze River Delta (YRD).

9. Lines 14 – 29: the font size of Abstract does not seem consistent; lines 14-19 font size seems smaller than those of lines 20-29.

**Response:** We thank the reviewer for pointing out the formatting issue in the Abstract. The inconsistency in font size between lines 14–19 and lines 20–29 resulted from a formatting artifact during manuscript preparation. We have now corrected the font settings so that the entire Abstract is presented in a uniform and journal-compliant font size in the revised version.

*We would like to once again express our sincere gratitude to the reviewers for their thoughtful and constructive comments. Their insights have been invaluable and have greatly enhanced the clarity, rigor, and overall quality of our manuscript.*